# Deletion of the Autism-Associated Protein SHANK3 Abolishes Structural Synaptic Plasticity after Brain Trauma

**DOI:** 10.3390/ijms23116081

**Published:** 2022-05-29

**Authors:** Carolina Urrutia-Ruiz, Daniel Rombach, Silvia Cursano, Susanne Gerlach-Arbeiter, Michael Schoen, Juergen Bockmann, Maria Demestre, Tobias M. Boeckers

**Affiliations:** 1Institute for Anatomy and Cell Biology, Albert Einstein Allee 11, 89081 Ulm, Germany; carolina.urrutia-ruiz@uni-ulm.de (C.U.-R.); daniel.rombach@uni-ulm.de (D.R.); silvia.cursano@uni-ulm.de (S.C.); susanne.gerlach-arbeiter@uni-ulm.de (S.G.-A.); michael.schoen@uni-ulm.de (M.S.); juergen.bockmann@uni-ulm.de (J.B.); maria.demestre@uni-ulm.de (M.D.); 2Deutsches Zentrum für Neurodegenerative Erkrankungen (DZNE), Translational Biochemistry, 89081 Ulm, Germany

**Keywords:** ASD, TBI, SHANK3, trauma, synapses, plasticity

## Abstract

Autism spectrum disorders (ASDs) are characterized by repetitive behaviors and impairments of sociability and communication. About 1% of ASD cases are caused by mutations of *SHANK3*, a major scaffolding protein of the postsynaptic density. We studied the role of SHANK3 in plastic changes of excitatory synapses within the central nervous system by employing mild traumatic brain injury (mTBI) in WT and *Shank3* knockout mice. In WT mice, mTBI triggered ipsi- and contralateral loss of hippocampal dendritic spines and excitatory synapses with a partial recovery over time. In contrast, no significant synaptic alterations were detected in *Shank3*∆*11−/−* mice, which showed fewer dendritic spines and excitatory synapses at baseline. In line, mTBI induced the upregulation of synaptic plasticity-related proteins Arc and p-cofilin only in WT mice. Interestingly, microglia proliferation was observed in WT mice after mTBI but not in *Shank3*∆*11−/−* mice. Finally, we detected TBI-induced increased fear memory at the behavioral level, whereas in *Shank3*∆*11−/−* animals, the already-enhanced fear memory levels increased only slightly after mTBI. Our data show the lack of structural synaptic plasticity in *Shank3* knockout mice that might explain at least in part the rigidity of behaviors, problems in adjusting to new situations and cognitive deficits seen in ASDs.

## 1. Introduction

Autism Spectrum Disorders (ASDs) are a group of neurodevelopmental syndromes mainly characterized by the inability to socialize, communication impairments, repetitive behaviors and behavioral rigidity [1,2]. Some ASDs have been consistently linked to genetic mutations in genes encoding proteins vital for a proper synaptic structure and function, such as the *SHANK3* gene [3]. Indeed, ∼1% of the ASD cases are caused by mutations or haploinsufficiency in the *SHANK3* gene [4], the first isoform described to cause ASD among the three ProSAP/Shank protein family members [5]. SHANK3 is a scaffolding protein that localizes to the core of postsynaptic densities (PSDs) at excitatory synapses. The PSD structure serves as an essential scaffold that organizes and anchors glutamate receptors, signaling molecules and actin filaments that contribute to the proper synapses’ structure and functionality [6]. Via a whole series of different protein-interaction domains, SHANK3 interconnects several proteins within the PSD with actin-binding proteins and can be considered a “master organizer” protein of the PSD [6]. Indeed, SHANK3 attaches at different sites to the actin cytoskeleton, which controls the dendritic spine shape by dynamic remodeling. Secondary to this, it also alters its functionality [7]. In this respect, it should be mentioned that SHANK3 overexpression is sufficient to induce the generation of new dendritic spines in aspiny cerebellar neurons [8], indicating the potential of SHANK3 to modulate the structure of synaptic contacts effectively. 

In addition to this, the generation and analysis of mice lines lacking different SHANK3 isoforms have revealed insights into SHANK3 relevance at the synapses. At the morphological level, these animals showed reduced dendritic spine density with longer spines and no alterations in the PSD ultrastructure [9], while others exhibited reduced dendritic spine density and smaller PSD [10,11]. Furthermore, the PSDs of these animals are known to have fewer proteins responsible for local actin remodeling as well as other synaptic proteins, such as Homer [10,12,13,14,15,16]. At the behavioral level, *Shank3*-deficient animals display several ASD core symptoms, such as repetitive behavior, self-grooming [10,13,14,15,17,18,19] and social novelty deficits [10,16,19], along with ASD comorbidities, such as slight impairments in learning [13,14,17,19] and cognitive rigidity [15,19]. Moreover, most of these *Shank3* knockout animal models have confirmed a decisive role for SHANK3 in synaptic plasticity with high consistency. For instance, *Shank3* knockout mice present reduced basal glutamatergic synaptic transmission [9,17,19], hippocampal long-term potentiation impairments [9,13,14,16,17,19] and loss of intrinsic homeostatic plasticity, which is known to counteract maladaptive effects of long-term plasticity [20], providing, thus, compelling evidence that SHANK3 expression levels are essential for the normal function of the defined brain circuits. Opposite results have been observed regarding the effects of *Shank3* deletion on the presynaptic side, where SHANK3 is also present at least at early developmental time points [21]. For instance, the presynaptic release in D2-medium spiny neurons was reduced [22] or increased in heterozygous mice [9], or with no changes in the hippocampal presynaptic release of *Shank3* knockout mice [19,23]. Additionally, SHANK3 is not only expressed in excitatory but in inhibitory neurons. GABAergic neurons in the prefrontal cortex and dorsolateral striatum express SHANK3. *Shank3* specific deletion in GABAergic neurons suppressed excitatory transmission without impacting inhibitory transmission; moreover, the abnormal social and locomotor behavior of the global *Shank3* knockout mice in this study was replicated in the *Shank3*-GABAergic knockout mice [24].

To investigate if and how *Shank3*-deficient animals are able to respond to a physical impact on brain structures, we employed a mild traumatic brain injury (mTBI) paradigm to animals lacking some of the major SHANK3 isoforms (*Shank3*∆*11−/−* mice) and to wild-type mice. The mTBI paradigm is already known to induce an immediate loss followed by a time-dependent partial recovery of hippocampal synapses, indicating a fast and widespread form of synaptic plasticity [25,26,27]. This system also enabled us to visualize how synapses would react to an outer impact on the ipsilateral and contralateral sides of the injury. Moreover, we could assess whether an mTBI serves as a two-hit model for ASD in *Shank3*∆*11−/−* mice by assessing behavioral alterations that typically are found in animal models for ASD [28]. We focused on the hippocampus, where SHANK3 is mainly enriched [21,29], and it is known to play a role in long-term hippocampal potentiation, as mentioned above. On the other hand, after an mTBI, a fifth of the patients do not fully recover and show signs of memory impairment without a distinguishable injury [30]. Therefore, we also studied trace-fear memory protocol dependent on hippocampal function [31] and analyzed how fear memory is affected by mTBI in both genotypes.

We found that SHANK3 is essential to induce synaptic plasticity after an mTBI confirming that *Shank3*∆*11−/−* mice present structural synaptic plasticity impairments, and we found that behavior was subtly affected in both strains following mTBI. Additionally, some of the proteins known to mediate crucial structural and functional plasticity changes following mTBI are dysregulated in *Shank3*-deficient mice. To our knowledge, this is the first time that brain trauma has been inflicted on a mice model of autism. In summary, we found good evidence that SHANK3 isoforms are essential for plastic changes of excitatory synapses. In turn, *Shank3* knockout mice completely lack structural synaptic plasticity, which might at least in part explain the rigidity of behaviors, problems in adjusting to new situations and cognitive deficits seen in ASDs.

## 2. Results

### 2.1. Mild Traumatic Brain Injury (mTBI) Causes Excitatory Synapses Loss Only in Wild-Type but Not in Shank3∆11−/− Mice

We analyzed the dendritic spine density of excitatory synapses in the hippocampal CA1 pyramidal neurons of both WT and *Shank3*∆*11−/−* mice after 5, 10, 18 and 30 days following mTBI (Figure 1a). At baseline, WT animals presented more dendritic spines than *Shank3*∆*11−/−* mice. After mTBI, the spine density in WT animals presented a considerable reduction on the ipsilateral (down to 40%) and contralateral sides (down to 30%) of the injury with no recovery at 30 dpi (days post-injury). However, *Shank3*∆*11−/−*, which presented at baseline with fewer dendritic spines, did not lose dendritic spines after the time points assessed on both ipsilateral and contralateral sides (Figure 1b), showing, thus, no response or reaction to the injury at the dendritic spine density level. Next, we rated the spine maturation dynamics by classifying the spine morphology from less to more mature (filopodia, long-thin, thin, stubby, mushrooms and branched spines). At baseline, WT mice presented predominantly mushroom and stubby and fewer filopodia spines. After mTBI, WT mice displayed an increase in filopodia and a decrease in the more mature stubby, mushroom and branched spines after mTBI, whereas *Shank3*∆*11−/−*, which at baseline presented more filopodia and few mature spines (stubby, mushroom and branched) than WT animals, exhibited after mTBI a loss in filopodia, an increase in long-thin, a decrease of stubby and no changes of more mature mushroom and branched spines (Figure 1c). Additionally, we quantified the number of excitatory synapses at the protein level by immunostaining, assessing the amount of co-localization between the postsynaptic protein PSD-95 and the presynaptic marker piccolo. We observed that the spine reduction in WT animals appeared along with a loss of these excitatory markers, presenting fewer co-localization in the CA1 and CA3 hippocampus in both ipsilateral and contralateral sides after mTBI. While *Shank3*∆*11−/−* mice, which at baseline presented fewer PSD-95/Piccolo positive excitatory synapses, exhibited a slight, albeit non-statistically significant reduction of the synaptic contacts after mTBI in both the ipsilateral and contralateral sides (Figure 1d), stating that *Shank3*∆*11−/−* mice do not react dramatically to the mTBI at the synaptic level.

### 2.2. The PSD’s Ultrastructure Recovers Mostly in Wild-Type but Not Shank3∆11−/− Animals after mTBI

In line with the prior findings, the analysis of excitatory synapses (with thick and known mesh-like postsynaptic density structures) of the CA1 hippocampus by transmission electron microscopy (TEM) showed a statistically significant reduction in the amount of PSDs structures in WT animals at the different time points evaluated on both the ipsilateral and contralateral sides of the CA1 hippocampus with no recovery. In contrast, *Shank3*∆*11−/−* mice presented at baseline fewer PSD, with no changes in the number of PSDs following mTBI (Figure 2a,b). Regarding the PSD ultrastructure, mTBI induced a drop in the PSD length between five and 18 dpi, which recovered close to the baseline levels after 30 dpi on the ipsilateral CA1 hippocampus. However, the PSD thickness did not recover at 30 days in WT animals on the ipsilateral side (Figure 2b). In contrast, at baseline, *Shank3*∆*11−/−* mice exhibited significantly shorter PSD than WT mice but with similar thickness between both genotypes on the ipsilateral side.

After mTBI, the PSD length remained unaltered in *Shank3*∆*11−/−* mice, but the PSD thickness was slightly thinner after ten dpi on the ipsilateral CA1 hippocampus (Figure 2c). In addition, the mathematical calculation of the synapse volume (V = (length/2)^2^ × thickness × π) showed a reduction with partial recovery at 30 days in WT animals on the ipsilateral side, while at baseline, *Shank3*∆*11−/−* mice already exhibited a smaller synapse volume that remained unaltered by mTBI (Figure 2d). Moreover, we did not find statistically significant changes at the synaptic ultrastructural level in WT mice or *Shank3*∆*11−/−* mice on the contralateral side following mTBI.

### 2.3. mTBI Causes No Neuronal Loss Along with Astrocyte Proliferation at the Hippocampus in Wild-Type and Shank3∆11−/− Mice

To determine whether the dendritic spines and synapse loss observed after mTBI was due to neuronal loss in the hippocampus, we stained for the neuronal marker NeuN in the CA1 and CA3 regions. No neuronal loss was found in both strains, ruling out the hypothesis that neuronal degeneration causes synaptic loss in the hippocampus. Nevertheless, in other brain regions, such as the cerebral cortex, both WT and *Shank3*∆*11−/−* mice presented neuronal loss after mTBI on the ipsilateral side (Figure 3a). Likewise, we wanted to assess how the glia reacted to the brain injury. Therefore, we analyzed the activation of the astrocytes by quantifying the number of GFAP positive cells in the CA1 and CA3 hippocampus. We found in the CA1 hippocampal region an increase in the number of GFAP positive cells on the ipsilateral side in both genotypes at ten dpi, but on the contralateral side, we saw that the astrocyte number tended to augment in WT mice and increased statistically significant only in *Shank3*∆*11−/−* after ten dpi. However, in the CA3 region, there was only a statistically significant increase in the astrocyte number in WT but not in *Shank3*∆*11−/−* mice. Moreover, this response was observed both on the ipsilateral and contralateral sides after ten dpi (Figure 3b).

### 2.4. Microglia Are Activated in Both Wild-Type and Shank3∆11−/− Animals, but Proliferate Only in WT Animals Following mTBI

We assessed the involvement of local inflammation by the histological analysis of microglia proliferation, counting the number of positive Iba-1 cells. Following mTBI, we saw increased microglia after ten dpi in WT animals on the ipsilateral and contralateral sides of the CA1 hippocampus. Nevertheless, we did not observe statistically significant changes in the microglia number on the ipsilateral and contralateral sides of the CA3 hippocampus of WT animals after the injury. In contrast, *Shank3*∆*11−/−* animals did not exhibit overall microglial proliferation in the CA1 and CA3 hippocampus (Figure 4a). In addition, we evaluated whether mTBI induced activation of the remaining microglia. We evaluated the degree of microglial activation by exploring changes in microglia morphology. Three categories were considered: the resting state (highly ramified), the withdrawn stage (retraction of extant branches) and the motile stage (complete branches retraction, the activated one) [32]. We observed that WT and *Shank3*∆*11−/−* animals presented a similar degree of activation in CA1 and CA3 hippocampus on the ipsilateral side with a decrease in resting microglia; a similar increase in the number of withdrawn and motile microglia after the different time points analyzed. Moreover, regarding the microglial activation on the contralateral side, we found that the degree of microglial activation was slightly earlier in *Shank3*∆*11−/−* mice compared to WT mice (Figure 4b).

### 2.5. Screening for Potential Mechanism(s) Accounting for Synaptic Loss Reveals a Potential Role for Proteins That Participates in Functional and Structural Synaptic Plasticity Following mTBI

We recently demonstrated that the upregulation of the Corticotropin-Releasing Hormone (CRH) in the hippocampus accounted for synapses loss in a chest trauma model [32]. Therefore, we tested this hormone in our mTBI model and saw that both strains showed similar levels at baseline in the CA1 and CA3 hippocampus. However, only *Shank3*∆*11−/−* mice showed upregulation due to mTBI in the CA1 ipsilateral hippocampus, while WT animals only presented a minor increase at 5 dpi, although not statistically significant (Appendix A).

Thus far, we have seen that WT animals presented some degree of synaptic plasticity demonstrated by the following findings: (i) the dendritic spines showed spine remodeling assessed by an increase of the filopodia spines that may give rise to more mature spines at later points, (ii) the excitatory synapses in the CA1 region recovered partially at later time points and (iii) the PSD ultrastructure exhibited a recovery in length and volume at 30 dpi. In contrast, our *Shank3*∆*11−/−* mice did not lose excitatory synapses and showed perturbed dendritic spine structural remodeling following mTBI. These findings led us to hypothesize that *Shank3*∆*11−/−* mice display impairments in morphological plasticity. First, in *Shank3*∆*11−/−* mice, mTBI induced a drop in the number of immature filopodia spines that should generate more mature spines; second, there was no statistically significant loss in the excitatory synapses number after mTBI; third, in the PSD ultrastructure, the length and volume remain barely unaltered after mTBI. Thus, we performed hippocampus protein fractionation of the ipsilateral side of the injury to search for changes at the protein level regarding proteins localized in the PSD structure and proteins that participate in functional and structural synaptic plasticity (Appendix A). First, we analyzed in crude synaptosomal fractions (P2) the expression of SHANK3, NR1 (NMDA receptor subunit 1), Abi-1 and Synaptic Ras GTPase (SynGAP) (Appendix A). Then, we evaluated the expression of proteins localized at the synapses such as SHANK2, GluA2, Homer b/c, mGluR5, GluA1 and synaptophysin, and finally, pCREB/CREB expression in the cytosolic fraction (S2) (Appendix A). We observed comparable levels of these proteins regardless of the genotype or mTBI, except for GluA1 (genotype *p* = 0.0192). Therefore, we focused on functional plasticity by analyzing the activity-regulated cytoskeletal protein Arc expression required for long-term synaptic plasticity and memory consolidation [33]. The Arc appeared more upregulated in WT than in *Shank3*∆*11−/−* sham mice (genotype *p* < 0.0001) (Figure 5b). We observed a decrease in Arc expression by immunostaining in *Shank3*∆*11−/−* compared to WT animals at ten dpi in the CA3 ipsilateral hippocampus (Figure 5a). We also evaluated essential proteins involved in the actin filament dynamics and the cytoskeletal reorganization that contribute to structural plasticity, such as drebrin A and cofilin [34]. For this, we used S2 fractions after performing screening in different subcellular fractions of hippocampi homogenates (Appendix A). First, we evaluated drebrin A expression given its importance as an actin-binding protein and as an F-actin stabilizer in the dendritic spines. In sham mice, drebrin A appears to be more expressed in WT than *Shank3*∆*11−/−* animals (genotype *p* < 0.05). After 30 dpi, we saw a decrease in the drebrin A levels in *Shank3*∆*11−/−* compared to WT mice counterparts (Figure 5c). Furthermore, we also analyzed the expression of proteins involved in actin polymerization. Cofilin is known to depolymerize F-actin due to its severing-actin activity, and the LIMK-1 kinase inhibits cofilin activity by phosphorylation. Moreover, phosphorylated cofilin is known to stabilize F-actin [34]. In sham mice, phosphorylated cofilin and cofilin expression did not differ between strains. Following mTBI, we only found a statistically significant difference at 30 dpi, seeing an increase in the levels of phosphorylated cofilin in WT animals but not in *Shank3*∆*11−/−* mice (Figure 5c).

### 2.6. Wild-Type and Shank3∆11−/− Mice Presented Moderate Behavioral Alterations after mTBI

We determined whether the subtle response observed in *Shank3*∆*11−/−* compared to WT mice regarding morphological and molecular changes may translate into specific behavioral changes after mTBI. Considering anxiety as an associated feature of ASD patients, we tested first whether mice presented anxiety-related behavior before and after mTBI. Being aware of the typical aversion that mice present to elevated areas, we conducted the elevated plus maze (EPM). At baseline, *Shank3*∆*11−/−* showed anxiety compared to WT mice. An mTBI did not induce anxiety at 30 dpi in WT mice only a trend to show fewer entries into the open arms. On the other hand, *Shank3*-deficient animals showed anxiety independent of the injury (Figure 6a). Second, we also evaluated anxiety using the open field test, where anxious animals spend more time on the arena’s borders. At baseline, *Shank3*∆*11−/−* sham mice spent less time in the arena center than WT sham animals (Appendix A), indicating that these animals present anxious behavior. After five days of mTBI, we observed that WT mice spent more time in the center of the arena. In contrast, *Shank3*∆*11−/−* did not alter its baseline behavior due to mTBI, spending more time on the arena’s borders (Appendix A). After 30 dpi, WT mice showed no alterations, and the knockout animals continued presenting anxiety in the open field regardless of mTBI (Appendix A). The assessment of locomotor activity in this test showed that at baseline, *Shank3*∆*11−/−* sham animals move less and are slower than WT animals. Moreover, an mTBI neither affected the locomotor activity of WT nor *Shank3*-deficient animals (Appendix A).

We also evaluated fear memory using a trace fear conditioning protocol that relies on hippocampal function after 5 and 30 days of mTBI [31]. When the animals learned the association between a tone and an electric foot shock in the acquisition phase, *Shank3*∆*11−/−* sham animals displayed a slightly enhanced fear memory learning (freezing 35% of the trial) compared to WT sham animals (freezing 20% of the trial) at 30 dpi. After five days of mTBI, we observed an increase in the freezing response in WT animals at the end of the trial (Figure 7c, upper panel). On the contrary, *Shank3*∆*11−/−* mice presented an enhanced freezing response independent of mTBI (30 dpi). Furthermore, the evaluation of the auditory memory showed that, again, *Shank3*∆*11−/−* exhibited an enhanced freezing behavior at baseline (30 dpi). Following mTBI, we did not observe changes in auditory memory in both genotypes (Figure 7c, middle panel). Finally, the evaluation of the contextual memory showed at baseline again an enhanced freezing response in *Shank3*∆*11−/−* mice with a minor increase in fear memory following 30 days of mTBI, while WT animals presented a tendency to show impairments in fear memory after 30 dpi, although not statistically significant (Figure 7c, bottom panel).

We were also interested in assessing whether trauma worsens the behavioral phenotype of *Shank3*∆*11−/−* and assessed another core feature of ASD; thus, we evaluated repetitive behavior by quantifying the animal’s time doing self-grooming. At the basal level, *Shank3*∆*11−/−* sham animals spent most of the time performing self-grooming, and mTBI did not induce self-grooming in WT animals or worsen the overgroom phenotype of *Shank3*∆*11−/−* mice (Figure 6b).

Finally, as a summary, we outlined the main conclusions obtained in this study in a graphical illustration (Figure 8). In WT mice, distinct loss and incomplete recovery dynamics can be seen at synapses respecting dendritic spine number, excitatory synapses and the ultrastructural level of the PSD in the CA1 hippocampus. In contrast, *Shank3*-deficient animals exhibited no apparent alterations respecting dendritic spine number and PSD ultrastructure and subtle alterations at the level of excitatory synapses (Figure 8a). Furthermore, at the cellular level, we observed proliferation of astrocytes in both strains following mTBI, whereas the microglia increased in WT and did not proliferate in *Shank3*∆*11−/−* mice following ten dpi in the CA1 hippocampus and with no loss of hippocampal neurons (Figure 8b). Concerning the proteins that participate at the level of structural and functional synaptic plasticity, we saw a pronounced increase of drebrin A and p-cofilin, both functioning as F-actin stabilizers in WT animals following mTBI compared to *Shank3*-deficient animals and an evident lower expression of Arc in knockout animals compared to the WT counterparts following mTBI (Figure 8c). Finally, the behavioral assessment showed that *Shank3*∆*11−/−* presented enhanced anxiety and exacerbated self-grooming, behaviors that mTBI subtly modified. At 30 dpi, *Shank3*∆*11−/−* showed increased fear memory after mTBI. In contrast, WT mice exhibited increased freezing behavior at 5 dpi and a tendency to be anxious at 30 dpi, demonstrating a behavioral reaction to the brain lesion that we slightly observed in *Shank3*-deficient animals (Figure 8d).

## 3. Discussion

An important observation emerges from our study. *Shank3*∆*11−/−* mice show subtle synaptic response to a physical brain impact induced by mTBI. Research has shown that mutations in *SHANK3* or its haploinsufficiency account for at least ∼1% of ASD cases [4], in which individuals present social and communication impairments, repetitive behavior and rigid behavior patterns. In the present study, we used the mTBI paradigm—known to induce the loss and partial recovery of synapses in time [26,27]—to assess whether the plastic changes that lead to the loss and recovery of synapses are dependent on the SHANK3 expression levels. Thus, we performed mTBI in animals lacking some of the major SHANK3 isoforms, the *Shank3*∆*11−/−* mice and in WT mice and determined whether structural synaptic plasticity is impaired in the *Shank3*∆*11−/−* mice.

We found that *Shank3*∆*11−/−* mice have fewer dendritic spines with altered morphology, fewer excitatory synapses and smaller PSDs (Figure 1 and Figure 2). These changes indicate profound changes at the synapse, where SHANK3 is predominantly located [6]. Therefore, *Shank3* knockout animals support these findings, exhibiting dendritic spine remodeling impairments [9,17], decreased spine density [10,13], longer dendritic spines and unaltered [13] or smaller PSDs [10]. Following mTBI, we observed in the WT-mice’s hippocampus loss of dendritic spines [26] along with excitatory synapses [35], which dropped substantially and reached a stable number at 30 days, but not a complete recovery (Figure 1).

In contrast, *Shank3*∆*11−/−* mice exhibited no dendritic spines loss, only minor changes in dendritic spine morphology, slight loss of excitatory synapses and no changes in the number of PSD structures, indicating a subtle response after mTBI at the synaptic level. Concerning the morphological changes seen in dendritic spines following mTBI, WT animals showed a loss of more mature spines (mushroom and stubby spines), and an increase in filopodia spines, whereas *Shank3*∆*11−/−* mice exhibited a loss of the filopodia, an increase in long-thin spines and a later decrease of stubby and no alterations of mushroom spines. Filopodia spines have been regarded as “immature” [36] and able to participate in synaptogenesis [37]. Consequently, SHANK3 overexpression induced a decrease in the number of filopodia spines on hippocampal neurons [7], and SHANK3-knockdown induced longer spines [8]. Depending on the injury model used, the literature reports different outcomes regarding spine morphology after mTBI in WT mice. For example, a mild controlled cortical impact induced an increase in filopodia spines [38], and a moderate impact caused a decrease in filopodia and no alterations in the number of stubby spines [26]. With a more severe controlled cortical impact, mushroom and stubby spines diminished, and the filopodia number remained unchanged [39]. Hence, our results show that *Shank3* deficiency abrogates the effects of mTBI observed in WT animals. Thus, these results may indicate that the dendritic spine dynamics and morphological plasticity changes following mTBI rely directly or indirectly on SHANK3 expression levels [40]. Comparable results have been seen in mice deficient for the metalloproteinase 9, which presented no alterations in the number of dendritic spines or its morphology after a mild controlled cortical impact [38].

Additionally, when we assessed ultrastructural changes, WT animals exhibited a decrease in the number of PSDs [35,41] with no recovery at 30 dpi, while *Shank3*∆*11−/−* mice exhibited no loss, and the PSD length and volume was unchanged. WT mice, however, presented plastic changes of synaptic contacts following mTBI [42] (i.e., a decrease in the length and volume of the PSDs), which recovered at later time points (Figure 2). Importantly, we observed in WT mice that the loss of dendritic spines, excitatory synapses and PSD structures was found on the ipsilateral and contralateral side of the injury, indicating that the TBI effects are not restricted to the lesion site [43,44].

In a next step, we searched for mechanisms/factors that might explain our findings and we evaluated neuronal loss, astrocyte proliferation, microglial activation and CRH expression in the hippocampus. We did not observe a hippocampal neuronal loss in both genotypes, as already demonstrated in previous studies [21] (Figure 3a). As a reaction to the impact, we saw, however, an increase in astrocytes number on the ipsilateral side in the CA1 hippocampus [41,45] independent of *Shank3* loss (Figure 3b). Astrocytes also increased on the contralateral side of the hippocampus for knockout animals in the CA1 region and WT animals in the CA3 region. Concerning microglia, sham mice of both strains have a similar number at baseline [46]. Following mTBI, the microglia only proliferated in WT mice CA1 hippocampus on both the ipsilateral and contralateral sides. Even though microglia were increased only in WT mice after mTBI, we observed an increment in the activated (motile) and a reduction in the less activated (resting) microglia in both strains [47] on the ipsilateral side (Figure 4). The observation that astrocytes and microglia increased also on the contralateral side indicates that the augmented gliosis occurring in the ipsilateral side is also seen in distant regions and may explain the loss of synaptic contacts [33,34]. Finally, we also determined the CHR expression levels because a CRH upregulation in the hippocampus is known to be a stress response in the brain regulating synapse number. Here, we observed a comparable expression of CRH in both strains and only a slight upregulation following mTBI in *Shank3*∆*11−/−* animals (Appendix A), suggesting that at least in this model, CRH is not responsible for the loss of dendritic spines [32,48].

The formation of neuronal connections or their reinforcement is the structural substrate of learning and memory, necessary for synaptic plasticity [49]. To elucidate whether the differences in the expression of essential proteins that regulate functional/structural synaptic plasticity accounts for the slight response in *Shank3*∆*11−/−* compared with the active dynamic process of damage and reparation of synapses presented in WT animals following mTBI, we evaluated the expression of several synaptic proteins. We detected a stable expression of most of these proteins in both genotypes following mTBI (Appendix A). Additionally, we analyzed the expression of the activity-regulated cytoskeletal protein Arc, which is localized at dendrites, modulating the trafficking of AMPA glutamate receptors [50,51] and the consolidation of enduring synaptic plasticity and memory [33]. We found reduced Arc expression at baseline in synaptosomal fractions of *Shank3*∆*11−/−* compared to WT sham mice. Furthermore, we observed a consistent upregulation of Arc expression by immunostainings only in CA3 hippocampus in WT but not in *Shank3*∆*11−/−* mice following mTBI (Figure 5a,b). In line with these results, it has already been shown that *Shank3^+/^*^∆*C*^ heterozygous mice displayed lower Arc expression [52]. Furthermore, these mice presented high histone methyltransferases levels. Indeed, the histone methyltransferases inhibition or knockdown reverted the *Shank3^+/^*^∆*C*^ mice social deficits and reestablished the NMDAR-mediated synaptic function, effects that were dependent on reestablishing Arc expression [53]. It is worth mentioning that Arc pertains to the immediate-early gene family, which expression is activated immediately after neuronal activity [54]. We did not analyze Arc expression at earlier time points than five days. Thus, more studies are needed to analyze the expression of intermediate-early genes following mTBI at earlier time points in WT and *Shank3*-deficient animals.

According to the prominent role of SHANK3 in the structural and functional organization of the dendritic spines [7], proteomic analysis of hippocampal PSD fractions of *Shank3*∆*11−/−* mice has shown a decrease in several proteins responsible for the actin cytoskeleton remodeling (Abi1, Gelsolin or Profilin2) [12]. SHANK3 localizes on the tip of the actin filaments in dendritic spines boosting its polymerization and inducing bigger dendritic spines by increasing the amount of F-actin in the spine. Whereas mutations in the ankyrin domain of *SHANK3*, the domain is missing in the *Shank3*∆*11−/−* mice, which induced longer dendritic spines and less F-actin contents [7]. Additionally, in *Shank3*-overexpressing animals, SHANK3 bound the Arp2/3 complex directly, incrementing the F-actin contents in dendritic spines [55]. Regulation of F-actin contents is critical for the formation and elimination of spines required for synaptic plasticity [56]. Hence, we assessed cofilin expression, which promotes F-actin depolarization due to its severing activity. We found neither changes under basal conditions nor after mTBI in both strains. However, cofilin phosphorylation, which stabilizes F-actin in dendritic spines and increases after long-term potentiation [57], presented comparable basal expression in both strains, although it only increased in WT, not in *Shank3*∆*11−/−* mice after 30 dpi (Figure 5c). These results suggest that F-actin is more stabilized in WT than *Shank3*∆*11−/−* mice, and in the latter, F-actin might be more depolymerized and unable to carry out spine remodeling. In line with these results, *Shank3^+/^*^∆*C*^ heterozygous mice presented lower levels of p-cofilin, reduced F-actin and blocking cofilin rescued ASD-like behaviors in these mice [58]. Likewise, *Shank3e^4−9^* also displayed a reduction in p-cofilin and the LIMK1 kinase, which phosphorylates cofilin, and therefore, inhibits it [58]. Thus, these data support the concept that *Shank3* deficiency leads to an increased cofilin activity that may lead to actin depolymerization, affecting the dendritic spine remodeling. Moreover, we evaluated the expression of drebrin A, also acknowledged for stabilizing F-actin in dendritic spines [34]. At the basal level, *Shank3*∆*11−/−* mice presented a slight reduction in drebrin A expression. After 30 days of mTBI, drebrin A expression dropped in *Shank3*∆*11−/−* compared with WT animals, indicating that dendritic spines in *Shank3*∆*11−/−* mice might be less stable. It is, however, unknown whether SHANK3 interacts directly with drebrin A or indirectly via Homer at excitatory synapses [59].

A closed head Injury can induce cognitive impairment in humans, and in the case of an mTBI, these deficits do not resolve in 20% of the cases, with a subgroup of patients who do not fully recover [60]. In this study, we assessed *Shank3*∆*11−/−* and WT mice behavior following mTBI. First, we assessed anxiety and a trace fear-conditioning paradigm, given that TBI is known to induce anxiety behavior, cognitive impairments [61,62] and post-traumatic stress disorder [63]. At baseline, *Shank3*-deficient animals were more anxious than their WT mice counterparts. After 30 days of mTBI, WT mice presented a tendency to increase anxiety, but on the contrary, *Shank3*∆*11−/−* mice did not alter their anxiety levels (Figure 6a). To corroborate that anxiety was unchanged after mTBI, we also performed the open field and saw that Shank3-deficient mice preferred to spend more time on the arena border, and this behavior was not exacerbated with the mTBI intervention (Appendix A–c).

We also assessed hippocampal function by the trace fear conditioning paradigm, which relies on the intact function of the hippocampus [31]. At baseline, sham *Shank3*∆*11−/−* mice compared to WT mice showed an intensified fear memory in all the trial phases analyzed. During the training phase, WT mice following mTBI showed an increase in the fear memory at the end of the trial compared to knockout mice at 5 dpi (Figure 7c, upper panel), showing that mTBI augmented the fear memory in WT mice. It has been seen that an mTBI can induce an overall increase in fear memory in rats [64]. In fact, a post-traumatic stress disorder (PTSD) is experienced at least in 13% of mTBI civil patients [65]. Interestingly, *Shank3*-deficient mice continue showing an exacerbated fear response despite mTBI. In the contextual memory trial, WT mice tended to lose the fear memory at 30 days following mTBI, though not statistically significant [41]. In contrast, *Shank3*∆*11−/−* animals exhibited a strengthened fear response in the contextual phase with an increase due to mTBI (Figure 7c, middle panel). Finally, we could neither alter nor worsen the repetitive behavior in both strains following mTBI (Figure 6b). Thus, a mild response at the biochemical and molecular synaptic level following mTBI was reflected at the behavioral level in *Shank3*-deficient mice. We observed only moderate behavioral alterations in these mice following mTBI, suggesting rigidity of behaviors already confirmed in these mice [15]. In fact, plasticity impairments have also been observed in the case of fragile X syndrome, in which the loss of FMRP protein, an RNA binding protein that regulates translation and long-term changes in synaptic strength, leads to intellectual disability and, in some cases, to ASD [66]. Like *Shank3*∆*11−/−*, these animals present dendritic spines with an immature phenotype; however, in humans, there is an increase in dendritic spine density, a phenotype that has not been consistently replicated in knockout mice [67]. These animals present a disrupted synapse formation by showing multiple innervated spines, leading to abnormal synaptogenesis [68], and in humans, the severity of the de syndrome in terms of intellectual disability is inversely correlated with the FMRP expression levels [69]. Within *SHANK* mutations in humans, *SHANK3* mutations have been related to more severe intellectual disability among the other SHANKS isoforms [70].

To our knowledge, this is the first study assessing an mTBI in a *Shank3* model for autism. Our results support the observation that autism-associated SHANK3 protein isoforms seem to be critical in inducing synaptic plasticity following mTBI. This notion is in line with an already-attributed role of SHANK3 in several knockout animals in long-term and homeostatic plasticity [9,13,14,16,17,19,20]. Therefore, we provide further evidence for underlying mechanisms that might explain the rigidity of behaviors and cognitive inflexibility related to the loss of crucial proteins involved with synaptic plasticity.

## 4. Materials and Methods

### 4.1. Animals

*Shank3*Δ*11−/−* mice were generated as described previously [18]. *Shank3*Δ*11−/−, Shank3*Δ*11+/-* mice and wild-type littermates were male (8–10 weeks old, weighting 25 ± 1.4 g). The breeding was performed by mating heterozygous mice on a C57BL/6 background. Animal housing followed standard laboratory conditions (average temperature of 22 °C, with a 12:12 light:dark cycle, water and food access ad libitum). Animal experiments were performed in compliance with the guidelines for the welfare of experimental animals issued by the Federal Government of Germany and authorized by the review board of Baden Wuerttemberg (Regierungspraesidium Tübingen), permit number 1233.

### 4.2. Traumatic Brain Injury

We used a previously described mild traumatic brain injury (mTBI) model [71]. TBI procedure was always performed early in the morning. Briefly, animals received a subcutaneous injection of buprenorphine (0.05 mg/kg) 30 min before mTBI and were put under sevoflurane anesthesia (Sevorane^TM^, Abbott, Wiesbaden, Germany) 3.5% in 96.5% O_2_ for induction and maintained with 2.5% in 97.5% O_2_ at a continuous flow FiO2 of 0.8 L min^−1^. The scalp was shaved, and a midline incision over the skull was performed. The head was placed and manually fixed within the stereotaxic frame, and the skin was retracted, exposing the impact area. The impactor rod was situated in the center of the left parietal bone. A 120 g weight-drop from a 2.5 mm height was used to perform TBI. When spontaneous breathing was restored, the scalp was sutured with Ethilon II 4-0 (Ethicon; Johnson and Johnson). As a sham control group, the animals underwent a similar experimental approach but without receiving the weight-drop impact. As a painkiller, further doses of buprenorphine were injected at 8 h intervals for the following 48 h post-injury (dpi). The general health of the mice was checked every three hours during the first 72 h and twice a day for the following week. The general health conditions of the animals were reported in a score sheet specially designed for this procedure. The analyses were performed after 5, 10, 18 and 30 dpi.

### 4.3. Golgi Staining Procedures

Mice were euthanized using a sevoflurane overdose. Then, according to the manufacturer’s instructions, the brains were collected and prepared for the Golgi-Cox staining procedure (FD Rapid GolgiStain^TM^ Kit). Afterward, brains were cut in 150 μm coronal sections with a vibratome (Microm HM 650) and mounted on gelatin-coated slides.

### 4.4. Dendritic Spine Imaging

Neurons of the pyramidal layer of the CA1 hippocampal sub-region were used to quantify dendritic spine density and morphology. We analyzed the apical dendrites of the stratum radiatum. For that, the acquisition of the Golgi-stained dendritic branches and spines was attained using a BZ-9000 Fluorescence Microscope (KEYENCE Corp., Osaka, Japan). The Image Z-stacks had a 0.4 μm focal steps size over the entire thickness of each dendrite and were obtained using a ×100/1.4 oil lens and reconstructed afterward using the ImageJ software. We assessed nine tertiary dendrites per animal and analyzed three to six animals per group. The spine density was assessed as the number of dendritic spines found per 10 μm dendrite length. The dendritic spine morphology was identified manually using the ImageJ software and following the classification described in [72], rating the dendritic spines in filopodia, long-thin, thin, stubby, mushroom and branched spines.

### 4.5. Slice Preparations and Immunohistochemistry

An intraperitoneal injection of ketamine-xylazine (100 mg/kg and 16 mg/kg, solubilized in 0.9% NaCl) was used as anesthesia. The transcardial perfusion was carried out with phosphate cold saline buffer (PBS−/−, without magnesium and calcium) followed by 4% paraformaldehyde (PFA). Then, the brains were collected, post-fixated for 18 h in 4% PFA at 4 °C and cryoprotected for 48 h with a 30% sucrose in PBS−/− solution at 4 °C. Afterward, the brains were frozen with TissueTek (Sakura) and stored at −80 °C. The brains were cut in 40 μm coronal cryosections spanning the parietal cortex (area of the injury) and were blocked with Serum Bovine Albumin 3%/Triton X-100 0.3% in PBS for 2 h. Primary antibodies were incubated in the blocking solution for 48 h at 4 °C on an orbital shaker. Next, the sections were washed with PBS 1x 0.5% Tween-20 (3 × 30 min), and subsequently, the secondary antibody Alexa Fluor^®^ was incubated at RT for 2 h in blocking solution. After that, sections were washed with PBS 1x 0.5% Tween-20 (3 × 30 min) and then mounted with VectaMount (Vector Laboratories Burlingame, CA). Images of immunostained sections were acquired using a confocal microscope (Leica SPE confocal microscope, Wetzlar, Germany). For immunostaining, the following antibodies were obtained from commercial suppliers: GFAP (1:500, Synaptic System GmbH, Goettingen, Germany, #173011), NeuN (1:1000, Synaptic System GmbH, Goettingen, Germany, #266004), Iba1 (1:500 Synaptic System GmbH, Goettingen, Germany, #234003), PSD95 (1:300, Abcam, Cambridge, England, #ab2723), CRH (1:1000, Abcam, Cambridge, England, #ab8901), Arc (1:300, Abcam, Cambridge, England, #ab183183) and Piccolo (1:500, Synaptic System GmbH, Goettingen, Germany, #142104). For immunohistochemistry, highly cross-adsorbed Alexa Fluor^®^ 488, 594, 647 conjugated antibodies were used (all 1:500, Jackson ImmunoResearch, Pennsylvania, USA). At least 3–5 animals were processed and analyzed, for each experimental group or time point.

#### Immunohistochemical Analysis

For the analysis of neuronal loss, astrocyte and microglial proliferation images were acquired using a Leica SPE confocal microscope (40× objective). The *z*-axis was 10 μm at 1 μm intervals for NeuN, GFAP and Iba-1 stainings. The analysis was performed using the cell counter plugin of Fiji ImageJ (National Institute of Health, Bethesda, USA) and the number of cells was normalized by area in mm^2^.

To analyze the morphology of the Iba1 positive microglia, the cells were characterized in three different stages [32]: the resting microglia state with a cell body <4 μm ^2^ and with branches number >4; the withdrawn microglia state with a cell body ≥4 μm^2^ and branches number ≤4; and the motile microglia state with a cell body <5 μm^2^ and branches number ≤2. The analysis was performed using the cell counter plugin of Fiji ImageJ (National Institute of Health, Bethesda, MD, USA).

To quantify synapses (PSD-95 and Piccolo) and CRH stainings, the *z*-axis acquisition depth was 2 μm and the images were acquired at intervals 0.2 μm. Three images were captured using a Leica SPE confocal microscope (63× objective). The synapses were quantified in the hippocampus (AP −1.34, ML ±2, DV 1.25–2 until AP −2.3, ML ±3, DV 1.25–2.5) according to the stereotaxic atlas [73]. The synapse quantification was performed using the Bitplane Imaris Software. Three random areas were selected in each picture. After setting an intensity threshold, the puncta per channel were counted. Afterward, the spot colocalization tool was used to determine the total number of synapses.

For CRH expression analysis, the mean intensity (integrated intensity) of fluorescence was determined using Fiji ImageJ software. Due to the impossibility of doing all the stainings in parallel, a sham control group was always included for determining the relative CRH expression.

### 4.6. Transmission Electron Microscopy

Animals were anesthetized and then transcardially pre-rinsed with a solution of 0.5% of heparin in 0.9% of NaCl and subsequently fixated with a solution containing 2% PFA, 2.5% glutaraldehyde and 1% saccharose in 0.1 M Cacodylate buffer pH 7.4. After fixation, brains were collected and post-fixated in 2.5% glutaraldehyde and 1% saccharose in 0.1 M Cacodylate buffer pH 7.4 overnight. The next day, the brains were put for 1 h in 1% saccharose in 0.1 M Cacodylate buffer pH 7.4 and cut in 200 μm coronal sections with a vibratome. Using a stereomicroscope, the CA1 hippocampus was dissected in 1 mm^2^ section and stored in 0.1 M Cacodylate buffer pH 7.4. Afterward, the sections were washed with 0.1 M phosphate buffer saline, post-fixed with 2% osmium tetroxide and dehydrated using ascending propanol series (30% to 90%). Furthermore, specimens were contrasted using 2% uranyl acetate, and the material was embedded in epon. Then, the specimens were cut in 0.5 μm sections with a microtome and were stained with toluidine blue. The CA1 region of the hippocampus was further selected under a light microscope and cropped in the epon embedded pieces, cut in 70–80 nm ultrathin sections and collected on 300 mesh copper grids [18].

#### Transmission Electron Microscopy Analysis

The samples were examined using a transmission electron microscope (TEM) Jeol JEM 14,000 at 120 kV. A magnification of 25,000× was used to determine the number of postsynaptic densities structures (PDSs). Synapses were analyzed in the CA1 stratum radiatum region of the hippocampus in an area of 35.8 μm^2^, calculating the final PSDs number in mm^2^. To analyze the PSDs ultrastructure in terms of length and thickness, an 80,000× magnification was used. Only synapses with clearly identifiable PSDs (presynaptic vesicles and postsynaptic terminals) were analyzed. The measurements of the PSD length and thickness were determined using ImageJ software. To determine the PSD volume, the following formula was used: (length/2)2 × thickness × π [32]. In total, 40 PSDs were analyzed per animal. At least three animals were assessed for each experimental group or time point.

### 4.7. Subcellular Fractionation and Western Blotting

Mice were euthanized with a sevoflurane overdose. Then, the hippocampi brains were dissected, separating the ipsilateral and the contralateral hemispheres. Afterward, differential fractionation was performed as described previously (Appendix A), and we assessed the quality of the fractions by analyzing the expression of SHANK3, PSD-95 and synaptophysin (Appendix A) [12]. Briefly, hippocampi were homogenized with a Teflon douncer, applying 12 strokes at 900 rpm in a solution of 10 mM HEPES pH 7.4, 2 mM EDTA, 5 mM Sodium orthovanadate, 30 mM Sodium fluoride, 20 mM β-glycerolphosphate and a protease inhibitor cocktail (Roche). The homogenates obtained (Ho fractions) were further centrifuged at 500× *g* × 5 min at 4 °C to discard nuclei, extracellular matrix and cell debris. Then, the supernatant was further centrifuged at 10,000× *g* × 15 min at 4 °C to separate the crude synaptosomal pellets (P2 fraction) from the cytosolic compartment (S2 fraction). The P2 pellet was resuspended in 50 mM HEPES pH 7.4, 2 mM EDTA, 2 mM EGTA, 5 mM Sodium orthovanadate, 30 mM Sodium fluoride, 20 mM β- glycerolphosphate, 1% Triton X-100 and a protease inhibitor cocktail (Roche) and further centrifugated at 20,000× *g* × 80 min. After the centrifugation, the postsynaptic pellet fraction (P3 fraction) was obtained and the synaptic cytosol supernatant (S3 fraction). The P3 pellet was finally resuspended in 50 μL Buffer 3 (50 mM Tris pH 9, 5 mM Sodium orthovanadate, 30 mM Sodium fluoride, 20 mM β-glycerolphosphate, 1% NaDOC and a protease inhibitor cocktail (Roche)) and all the fractions were frozen in liquid nitrogen and stored at −80 °C.

To perform the Western blot, the tissue lysates were separated in 8–12% gels by SDS-PAGE. Gels were transferred to a Trans-Blot Turbo Midi Nitrocellulose membrane BioRad) and blocked with 5% Bovine Serum Albumin in TBST (TBS, 0.2% Tween-20) for 1 h at room temperature under gentle agitation. Then, the membranes were cut and incubated overnight at 4 °C with the corresponding primary antibody. After washing the membranes five times with TBST, the membranes were incubated with the corresponding horseradish peroxidase (HRP)-conjugated secondary antibody at room temperature for 1 h. Finally, the membranes were rewashed five times, and the bands were visualized with Clarity Western ECL substrate (BioRad) and the MicroChemi 4.2 machine. For densitometry analysis, ImageJ software was used (http://rsbweb.nih.gov/ij/index.html, accessed on 24 May 2018). Three animals were analyzed for each experimental group or time point. The results of each experimental replicates are shown in (Appendix A). For Western blots, a homemade polyclonal antibody against Shank2 (1:500, “ppI-SAM pabSA5192”) and Shank3 (1:1000, “Fr1+2”) were used [18]. The following commercial antibodies were purchased: β3-tubulin (1:250,000, Covance, Burlington, North Carolina, USA; #PRB-435P), GluA2 (1:1000, Synaptic System GmbH, Goettingen, Germany, #182111), mGluR5 (1:1000, Millipore, Burlington, Massachusetts, USA #AB5675), Synaptophysin (1:20,000, Abcam, Cambridge, England, #ab14692), Homer (1:1000, Synaptic System GmbH, Goettingen, Germany, #160022), SynGAP (1:1000, Cell Signaling, Massachusetts, USA, #3200), phospho-cofilin (1:1000, Cell Signaling, Massachusetts, USA, #3313), cofilin (1:1000, Cell Signaling, Massachusetts, USA, #5175), drebrin A (1:1000, Cell Signaling, Massachusetts, USA #12243), Abi-1 (1:1000, Cell Signaling, Massachusetts, USA, #39444), NR1 (1:1000, Sigma, St Luois, USA #G8913), Arc (1:1000, Abcam, Cambridge, England, # ab183183), phospho-CREB (1:1000, Abcam, Cambridge, England #ab32096) and CREB (1:500, Abcam, Cambridge, England, #ab32515). Secondary antibodies used for Western blots were HRP-conjugated antibodies (1:1000, Dako, Glostrup, Denmark).

### 4.8. Behavioral Tests

Before conducting any behavioral assessment, the animals were housed in the behavior testing room for at least five days. Before each test, the animals were allowed to habituate to the behavior room for at least one hour. The behavioral tests were conducted between 1 and 6 PM. All the devices were cleaned with 70% of ethanol between each trial. Additionally, only one test was performed per day. The behavioral assessments were performed either after 5 or 30 days following mTBI and in different groups of animals; therefore, each time point had its corresponding sham group.

#### 4.8.1. Open Field

Animals were placed on the arena center (45.5 × 45.5 × 39.5 cm) illuminated with 100 lx in the center for a total of 30 min. The locomotor activity was assessed by determining the distance traveled and the speed. Anxiety was also evaluated in this test by quantifying the time spent on the center or arena borders using the Viewer 3 software (BIOSERVE GmbH, Bonn, Germany).

#### 4.8.2. Elevated Plus Maze for Anxiety-like Behavior

Anxiety-related behavior was performed using the elevate plus maze. The elevated maze was located 60 cm above the floor and entails two open and two closed arms (50 × 50 × 16 cm wall) and a central platform (5 cm × 5 cm). The device has an illumination of 50 lx. Animals were located in the center platform facing toward the closed arms. The trial was conducted for 5 min, and the device was cleaned with 70% ethanol between each trial. The number of entries into the open and closed arms were quantified using Viewer 3 software (BIOSERVE GmbH, Bonn, Germany).

#### 4.8.3. Self-Grooming

Animals were placed individually inside a clean empty cage (26.5 cm length × 20 cm width × 14 cm height) under dim red light (10 lux) for a total of 30 min. The recording was conducted in a sound-proof chamber. Each mouse received a ten min habituation period in the test cage, and then they were assessed for the next 20 min with a stopwatch for the accumulated time spent in self-grooming all body regions. A blinded experimenter analyzed the videos.

#### 4.8.4. Fear Conditioning

Trace cued and contextual fear memory was conducted using the Fear Conditioning System 46103 from Noldus/Ugo Basile. The fear condition procedure was conducted in three days following the protocol described by Lugo et al., [31]. On the first day (acquisition phase), animals were placed in the testing chamber (25 × 17 × 17 cms) with a wired floor grid and were allowed to explore the new environment for 2 min. Then, the animals were exposed to the conditioned stimulus (CS), a 20 s tone (85 dB, 2700 Hz), followed 20 s later (trace period) by the unconditioned stimulus (UC), a mild electric foot shock (2 s, 0.5 mA). The CS-UC presentation was repeated five times after a 200 s inter-trial period. Once the training was completed, animals remained an extra min in the chamber. On the second day (trace cued memory testing), animals were located in the same chamber but with a different context: walls were changed, white plexiglass covered the wired floor grid, and a vanilla essence was put to change the chamber’s odor. After a baseline of 2 min, the animals were presented three times with the 20 s tone separated by an inter-trial period of 220 s between each tone presentation. On the third day (contextual test), the animals were placed in the same context as day one for 8 min without tone or electric foot shock presentation. The fear-related behavior was scored using the Ethovision 12 software (Noldus Information Technology, Wageningen, the Netherlands Inc.). Thus, the activity state tool of the software was used, and the freezing response during the whole trial, defined as the absence of detected movement except for breathing [74], was assessed.

### 4.9. Statistical Analysis

Data are shown as the mean ± standard error of the mean. At least three mice were used for immunohistochemistry and protein biochemistry in each group, and eight animals for behavior (except for the EPM analysis at 5 dpi). Most of the experiments (except behavior) were normalized to the mean value of the WT sham group.

To analyze the dendritic spine density in each genotype, we used a one-way ANOVA or Kruskal–Wallis test, if needed. To analyze the dendritic spine morphology, excitatory synapses, electron microscopy quantification, neuronal loss, astrocytes proliferation, microglia proliferation, CRH expression, Western blot and behavior, we performed a two-way ANOVA to assess the effect of the genotype, the mTBI intervention or their interaction. Finally, for the analysis of the fear conditioning at 1 min intervals, a three-way mixed ANOVA including repetitive measures was used; to determine the effect of the genotype, the mTBI intervention and the time or among all these three factors. In addition, Bonferroni’s multiple comparison post hoc test was conducted to determine significance among the groups. A *p*-value less than 0.05 was considered statistically significant. Data were analyzed with SPSS (Version 21 for MAC OS X) and GraphPad 7.0.

## Figures and Tables

**Figure 1 ijms-23-06081-f001:**
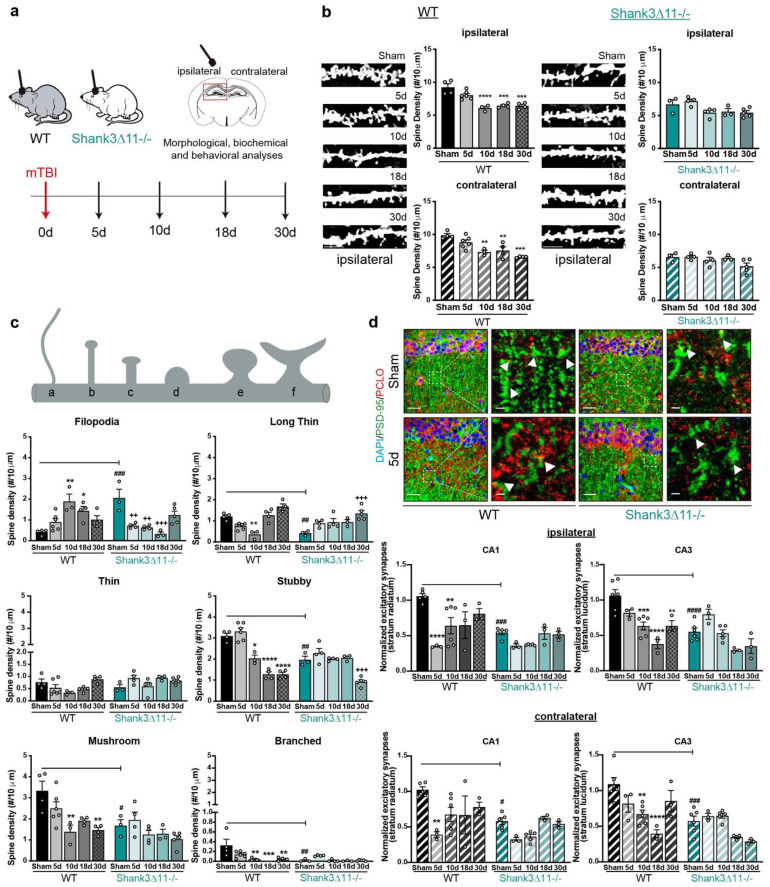
Wild-type mice but not *Shank3*-deficient animals exhibit a loss of excitatory synapses after mTBI. (**a**) Schematic representation of the experimental design of this study. (**b**) Analysis of the dendritic spine density in the CA1 hippocampus after Golgi staining. The number of spines was analyzed on the ipsilateral (filled bar) and contralateral (hatched bar) sides in WT (black bars, left panel) and *Shank3*∆*11−/−* animals (turquoise bars, right panel) after mTBI (scale bar, 5 μm). N = 3–6, error bars represent mean ± SEM; one-way ANOVA with Bonferroni’s post hoc comparison test was performed [WT ipsilateral side: F(4, 16) = 17.37, *p* = 0.0001; contralateral side: F(4, 16) = 11.63, *p* = 0.0001. *Shank3*∆*11−/−* ipsilateral side: F(4, 14) = 4.066, *p* = 0.0216; contralateral side Kruskal–Wallis test, *p* = 0.2553]. (**c**) Assessment of the spine morphology after mTBI. N = 3–6, error bars represent mean ± SEM; two-way ANOVA (genotype × mTBI) with Bonferroni’s post hoc comparison test was performed. [Filopodia spines: genotype F(1, 30) = 1.155, *p* = 0.2911; mTBI intervention F(4, 30) = 2.244, *p* = 0.0878; interaction F(4, 30) = 15.83, *p* = 0.0001]. Long-thin spines: genotype F(1, 30) = 3.603, *p* = 0.0673; mTBI intervention F(4, 30) = 17.86, *p* = 0.0001; interaction genotype × mTBI F(4, 30) = 8.855, *p* = 0.0001. Thin spines: genotype F(1, 30) = 5.610, *p* = 0.0245; mTBI intervention F(4, 30) = 3.108, *p* = 0.0297; interaction genotype × mTBI F(4, 30) = 3.226, *p* = 0.0257). Stubby spines: genotype F(1, 30) = 14.86, *p* = 0.0006; mTBI intervention F(4, 30) = 49.26, *p* = 0.0001; interaction genotype × mTBI F(4, 30) = 13.90, *p* = 0.0001. Mushroom spines: genotype F(1, 30) = 13.78, *p* = 0.0008; mTBI intervention F(4, 30) = 7.558, *p* = 0.0002; interaction genotype × mTBI F(4, 30) = 1.830, *p* = 0.1491. Branched spines: genotype F(1, 30) = 8.436, *p* = 0.0068; mTBI intervention F(4, 30) = 5.752, *p* = 0.0015; interaction genotype × mTBI F(4, 30) = 4.294, *p* = 0.0073]. (**d**) Immunohistochemical staining and quantification of excitatory synapses in the CA1 and CA3 hippocampus using the postsynaptic marker PSD-95 (green) and presynaptic marker piccolo (red), arrowheads indicate synapses (scale bar = 20 μm, magnification scale bar = 2 μm). Upper panel, representative images of the ipsilateral CA1 hippocampal region. Synapses were quantified on the ipsilateral (upper panel, filled bars) or the contralateral side (bottom panel, hatched bars) of the CA1 and CA3 hippocampus. N = 3–6, error bars represent mean ± SEM; two-way ANOVA (genotype × mTBI) with Bonferroni’s post hoc comparison test was performed [Ipsilateral side of the injury, CA1: genotype F(1, 29) = 19.53, *p* = 0.0001; mTBI intervention F(4, 29) = 8.597, *p* = 0.0001; interaction genotype × mTBI F(4, 29) = 2.940, *p* = 0.0372. CA3: genotype F(1, 31) = 18.96, *p* = 0.0001; mTBI intervention F(4, 31) = 15.46, *p* = 0.0001; interaction genotype × mTBI F(4, 31) = 4.812, *p* = 0.0039] [Contralateral side of the injury CA1: genotype F(1, 29) = 14.30, *p* = 0.0007; mTBI intervention F(4, 29) = 7.037, *p* = 0.0004; interaction genotype × mTBI F(4, 29) = 1.780, *p* = 0.1599. CA3: genotype F(1, 31) = 7.15, *p* < 0.0001; mTBI intervention F(4, 31) = 9.512, *p* < 0.0001; interaction genotype × mTBI F(4,31) = 5.881, *p* = 0.0012]. * *p* < 0.05, ** *p* < 0.01; *** *p* < 0.001; **** *p* < 0.0001: comparison regarding WT sham animals; ++ *p* < 0.01; +++ *p* < 0.001: comparison regarding *Shank3*∆*11−/−* sham animals; # *p* < 0.05, ## *p* < 0.01; ### *p* < 0.001; #### *p* < 0.0001: comparison between WT sham and *Shank3*∆*11−/−* sham animals.

**Figure 2 ijms-23-06081-f002:**
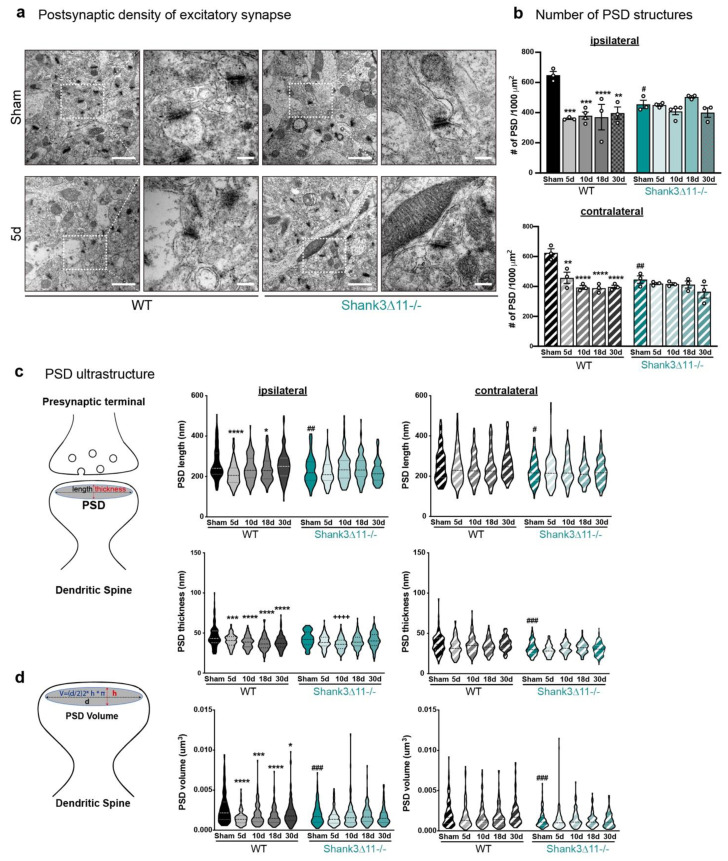
The overall number and ultrastructure of the postsynaptic density (PSD) of excitatory synapses are mainly affected in wild-type animals after mTBI. (**a**) Representative electron micrographs of synapses in the CA1 hippocampus ipsilateral side of WT mice (left panel) and *Shank3*∆*11−/−* (right panel) after 5 d of mTBI. White dashed squares represent the PSD structures quantified (with a 25,000× magnification: scale bar = 1000 nm), whose size were subsequently analyzed (80,000× magnification: scale bar = 200 nm). (**b**) Determination of the number of PSDs structures after mTBI on the ipsilateral side (upper panel, filled bars) and the contralateral side (bottom panel, hatched bars) for WT (black bars) and in *Shank3*∆*11−/−* mice (turquoise bars). N = 3, error bars represent mean ± SEM; two-way ANOVA (genotype × mTBI intervention) with Bonferroni’s post hoc comparison test was performed. [Ipsilateral side: genotype F(1, 22) = 0.3185, *p* = 0.5782; mTBI intervention F(4, 22) = 7.597, *p* = 0.0005; interaction genotype × mTBI F(4, 22) = 6.633, *p* = 0.0012. Contralateral side: genotype F(1, 20) = 6.916, *p* = 0.0161; mTBI intervention F(4, 20) = 12.89, *p* < 0.0001; interaction genotype × mTBI F(4, 20) = 5.836, *p* = 0.0028]. (**c**) Determination of the ultrastructure of the PSD. Assessment of the length and the thickness of the PSD structure, as depicted on the diagram. Measurements are represented on violin plots, where the median is depicted with dashed lines and the interquartile ranges in dotted lines. The length and thickness were measured on the ipsilateral (left panel, filled violin plots) and contralateral (right panel, hatched violin plots) sides of the CA1 hippocampus. N = 3. Data are shown as mean ± SEM; two-way ANOVA (genotype × mTBI intervention) with Bonferroni’s post hoc comparison test was performed. [PSD length on the ipsilateral side: genotype F(1, 1190) = 7.225, *p* = 0.0073; mTBI intervention F(4, 1190) = 8.094, *p* < 0.0001; interaction genotype × mTBI F(4, 1190) = 5.394, *p* = 0.0003. PSD length on the contralateral side: genotype F(1, 1190) = 36.99, *p* < 0.0001; mTBI intervention F(4, 1190) = 0.9649; interaction genotype × mTBI F(4, 1190) = 1.184; *p* = 0.3163]. [PSD thickness on the ipsilateral side: genotype F(1, 1190) = 0.6765, *p* = 0.4110; mTBI intervention F(4, 1190) = 19.09, *p* < 0.0001; interaction genotype × mTBI F(4, 1190) = 4.755, *p* = 0.0008. PSD thickness on the contralateral side: genotype F(1, 1190) = 63.37, *p* < 0.0001; mTBI intervention F(4, 1190) = 4.744, *p* = 0.0008; interaction genotype × mTBI F(4, 1190) = 2.692, *p* = 0.0298]. (**d**) Mathematical determination of the PSD volume as depicted on the diagram and represented in the violin plots, where the median is shown with dashed lines and the interquartile ranges with dotted lines. The PSD volume was determined on the ipsilateral (left panel, filled violin plot) and the contralateral sides (right panel, hatched violin plot) in both WT (black bars) and *Shank3*∆*11−/−* (turquoise bars) animals. N = 3, error bars represent mean ± SEM; two-way ANOVA (genotype × mTBI intervention) with Bonferroni’s post hoc comparison test was performed. [PSD volume on the ipsilateral side: genotype F(1, 1190) = 7.088, *p* = 0.0079; mTBI intervention F(4, 1190) = 11.57, *p* < 0.0001; interaction genotype × mTBI F(4, 1190) = 5.122, *p* = 0.0004. PSD volume on the contralateral side: genotype F(1, 1190) = 56.57, *p* < 0.0001; mTBI intervention F(4, 1190) = 1.468, *p* = 0.2096; interaction genotype × mTBI F(4, 1190) = 1.854, *p* = 0.1162]. * *p* < 0.05, ** *p* < 0.01; *** *p* < 0.001; **** *p* < 0.0001: comparison regarding WT sham animals; ++++ *p* < 0.0001: comparison regarding *Shank3*∆*11−/−* sham animals; # *p* < 0.05, ## *p* < 0.01; ### *p* < 0.001: comparison between WT sham and *Shank3*∆*11−/−* sham animals.

**Figure 3 ijms-23-06081-f003:**
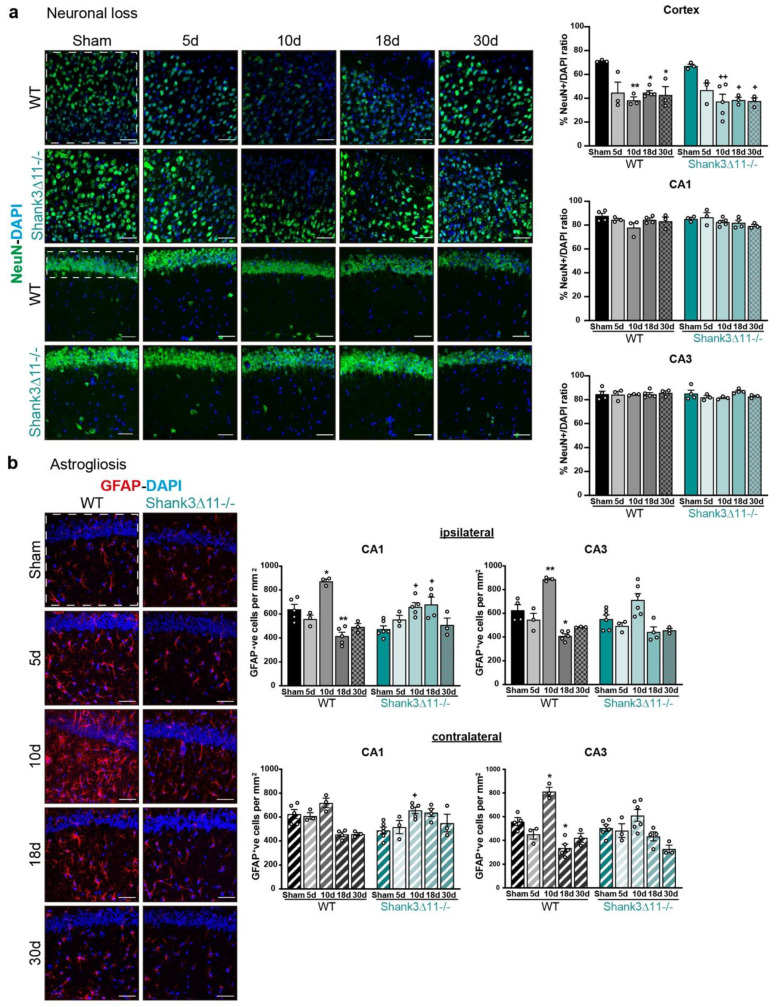
mTBI triggers no hippocampal neuronal loss and prompts astrocyte proliferation in both wild-type and *Shank3*∆*11−/−* mice. (**a**) Neurons were stained with the neuronal nuclear marker NeuN (green) and with DAPI (blue) on the brain ipsilateral cortex, CA1 and CA3 hippocampal region. Representative images (scale bar 50 μm) for the ipsilateral cortex (left first and second rows) and for the CA1 ipsilateral hippocampus (left third and fourth rows) of WT (first and third rows) and *Shank3*∆*11−/−* mice (second and fourth rows). The white dashed square/rectangle represents the region quantified. The NeuN percentage of positive nuclei quantification per field is shown for the ipsilateral cortex and the CA1 and CA3 ipsilateral hippocampus for WT (black bars) and *Shank3*∆*11−/−* mice (turquoise bars). N = 3–5. Data are shown as mean + SEM. Datasets were analyzed using two-way ANOVA (genotype × mTBI) with Bonferroni’s post hoc comparison test. [Ipsilateral cortex: genotype F(1, 24) = 0.8623, *p* = 0.3623; mTBI F(4, 24) = 12.85, *p* = 0.0001; genotype × mTBI F(4,24) = 0.2347, *p* = 0.9160. Ipsilateral CA1: genotype F(1, 25) = 0.09395, *p* = 0.7618; mTBI F(4, 25) = 2.619, *p* = 0.0591; genotype × mTBI F(1, 25) = 1.266, *p* = 0.3094. Ipsilateral CA3: genotype F(1, 25) = 0.7765, *p* = 0.3866; mTBI F(4, 25) = 0.9189, *p* = 0.4685; genotype × mTBI F(4, 25) = 0.9180, *p* = 0.4690]. (**b**) Astrocytes were stained with the astrocyte marker GFAP (red) and DAPI (blue). Representative images are shown for the different time points assessed in the CA1 hippocampal region following mTBI (scale bar 50 μm) (left panel). WT (left column) and *Shank3*∆*11−/−* mice (right column). The number of GFAP positive cells per mm^2^ was quantified on the ipsilateral side of the CA1 and CA3 hippocampus (white dashed square represents the region quantified) of WT mice (filled black bars) and *Shank3*∆*11−/−* mice (filled turquoise bars) and on the contralateral side of CA1 and CA3 hippocampus of WT (hatched black bars) or *Shank3*∆*11−/−* mice (hatched turquoise bars). N = 3–6. Data are shown as mean + SEM. Datasets were analyzed using two-way ANOVA (genotype × mTBI) with Bonferroni’s post hoc comparison test. [Ipsilateral CA1: genotype F(1, 30) = 0.6101, *p* = 0.4409; mTBI F(4, 30) = 12.95, *p* < 0.0001; genotype × mTBI F(4, 30) = 12.66, *p* < 0.0001. Ipsilateral CA3: genotype F(1, 30) = 4.848, p = 0.0355; mTBI F(4, 30) = 26.22, *p* < 0.0001; genotype × mTBI F(4, 30) = 1.838, *p* = 0.1476]. [Contralateral CA1: genotype F(1, 29) = 0.02442, *p* = 0.8769; mTBI F(4, 29) = 6.634, *p* = 0.0006; genotype × mTBI F(4, 29) 7.279, *p* = 0.0003. Contralateral CA3: genotype F(1, 30) = 2.684, *p* = 0.1118; mTBI F(4, 30) = 21.86, *p* < 0.0001; genotype × mTBI F(4, 30) = 3.998, *p* = 0.0102]. * *p* < 0.05, ** *p* < 0.01;: comparison regarding WT sham animals; + *p* < 0.05, ++ *p* < 0.01: comparison regarding *Shank3*∆*11−/−* sham animals.

**Figure 4 ijms-23-06081-f004:**
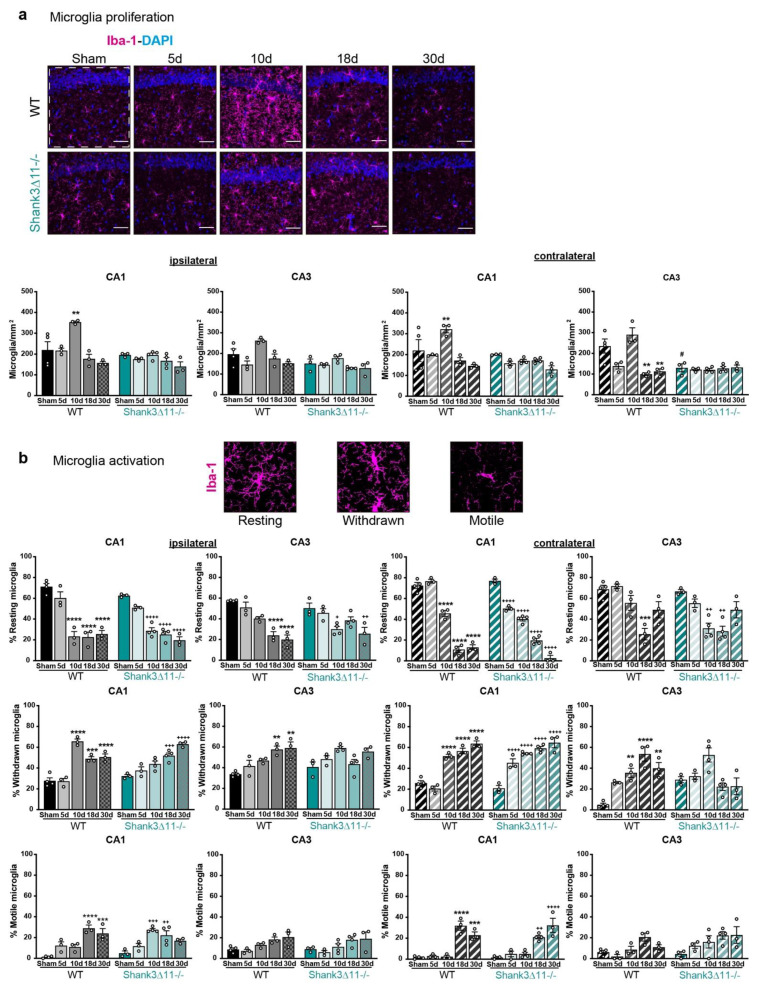
Microglial proliferation was only observed in WT animals, while its activation is similar in both wild-type and *Shank3*∆*11−/−* mice after mTBI. (**a**) Microglia were stained using the microglial marker Iba-1 (magenta) and nuclei were stained with DAPI (blue). Representative images are shown for each time point assessed for the CA1 hippocampal region on the left panel, WT mice (left column) and *Shank3*∆*11−/−* (right column) (scale bar 50 μm). The number of positive Iba-1 cells per mm^2^ (right panel) was quantified on the ipsilateral side of the CA1 and CA3 hippocampus (white dashed square represents the region quantified) of WT mice (filled black bars) and *Shank3*∆*11−/−* mice (filled turquoise bars) and on the contralateral side of CA1 and CA3 hippocampus of WT (hatched black bars) or *Shank3*∆*11−/−* mice (hatched turquoise bars). N = 3–6. Data are shown as mean + SEM. Datasets were analyzed using two-way ANOVA (genotype × mTBI), with Bonferroni’s post hoc comparison test. [Ipsilateral CA1: genotype F(1, 23) = 15.19, *p* = 0.0007; mTBI F(4, 23) = 11.15, *p* < 0.0001; genotype × mTBI F(4, 23) = 4.807, *p* = 0.0058. Ipsilateral CA3: genotype F(1, 23) = 15.53, *p* = 0.0007; mTBI F(4, 23) = 8.192, *p* = 0.0003; genotype × mTBI F(4, 23) = 1.859, *p* = 0.1521]. [Contralateral CA1: genotype F(1, 23) = 3.637, *p* = 0.0050; mTBI F(4, 23) = 6.335, *p* = 0.0014; genotype × mTBI F(4, 23) = 3.637, *p* = 0.0194. Contralateral CA3 genotype F(1, 23) = 16.22, *p* = 0.0005; mTBI F(4, 23) = 9.227, *p* = 0.0001; genotype × mTBI F(4, 23) = 10.13, *p* < 0.0001]. (**b**) Quantification of the microglial activation in percentage following mTBI into R: resting, W: withdrawn, M: motile microglia activation states in the CA1 and CA3 hippocampus for the ipsilateral side in WT mice (filled black bars) and *Shank3*∆*11−/−* mice (filled turquoise bars); and for the contralateral CA1 and CA3 hippocampus in WT mice (hatched black bars) and *Shank3*∆*11−/−* mice (hatched turquoise bars). Representative images of the microglial activation state are shown. N = 3–4. Data are shown as mean ± SEM. Datasets were analyzed using two-way ANOVA (genotype × mTBI) and Bonferroni’s post hoc comparison test. [CA1 ipsilateral resting: genotype F(1, 23) = 2.023, *p* = 0.1684; mTBI F(4, 23) = 69.03, *p* < 0.0001; genotype × mTBI F(4, 23) = 1.858, *p* = 0.1521. CA3 ipsilateral resting: genotype F(1, 23) = 0.05001, *p* = 0.8250; mTBI F(4, 23) = 22.33, *p* < 0.0001; genotype × mTBI F(4, 23) = 3.902, *p* = 0.0146]; [CA1 ipsilateral withdrawn: genotype F(1, 23) = 1.045, *p* = 0.3174; mTBI F(4, 23) = 45.71, *p* < 0.0001; genotype × mTBI F(4, 23) = 13.47, *p* < 0.0001. CA3 ipsilateral withdrawn: genotype F(1, 23) = 0.4755, *p* = 0.4974; mTBI F(4, 23) = 8.888, *p* = 0.0002; genotype × mTBI F(4, 23) = 4.123, *p* = 0.0116]. [CA1 ipsilateral motile: genotype F(1, 23) = 0.4149, *p* = 0.5259; mTBI F(4, 23) = 19.37, *p* < 0.0001; genotype × mTBI F(4, 23) = 5.887, *p* = 0.0021. CA3 ipsilateral motile: genotype F(1, 23) = 0.2422, *p* = 0.6273; mTBI F(4, 23) = 6.595, *p* = 0.0011; genotype × mTBI F(4, 23) = 0.9965, *p* = 0.0011]. [CA1 contralateral resting: genotype F(1, 23) = 14.79, *p* = 0.0008; mTBI F(4, 23) = 301.5, *p* < 0.0001; genotype × mTBI F(4, 23) = 16.52, *p* < 0.0001. CA3 contralateral resting: genotype F(1, 23) = 5.7000, *p* = 0.0256; mTBI F(4, 23) = 19.62, *p* < 0.0001; genotype × mTBI F(4, 23) = 2.536, *p* = 0.0677]; [CA1 contralateral withdrawn: genotype F(1, 23) = 9.657, *p* = 0.0050; mTBI F(4, 23) = 86.75, *p* < 0.0001; genotype × mTBI F(4, 23) = 8.869, *p* = 0.0002. CA3 contralateral withdrawn: genotype F(1, 23) = 0.01156 *p* = 0.9153; mTBI F(4, 23) = 9.152, *p* = 0.0001; genotype × mTBI F(4, 23) = 11.82, *p* < 0.0001]. [CA1 ipsilateral motile: genotype F(1, 23) = 0.1305, *p* = 0.7212; mTBI F(4, 23) = 45.26, *p* < 0.0001; genotype × mTBI F(4, 23) = 3.565, *p* = 0.0210. CA3 ipsilateral motile: genotype F(1, 23) = 4.960, *p* = 0.0360; mTBI F(4, 23) = 5.407, *p* = 0.0032; genotype × mTBI F(4, 23) = 0.9930, *p* = 0.4312]. ** *p* < 0.01; *** *p* < 0.001; **** *p* < 0.0001: comparison regarding WT sham animals; + *p* < 0.05, ++ *p* < 0.01; +++ *p* < 0.001; ++++ *p* < 0.0001: comparison regarding *Shank3*∆*11−/−* sham animals; # *p* < 0.05: comparison between WT sham and *Shank3*∆*11−/−* sham animals.

**Figure 5 ijms-23-06081-f005:**
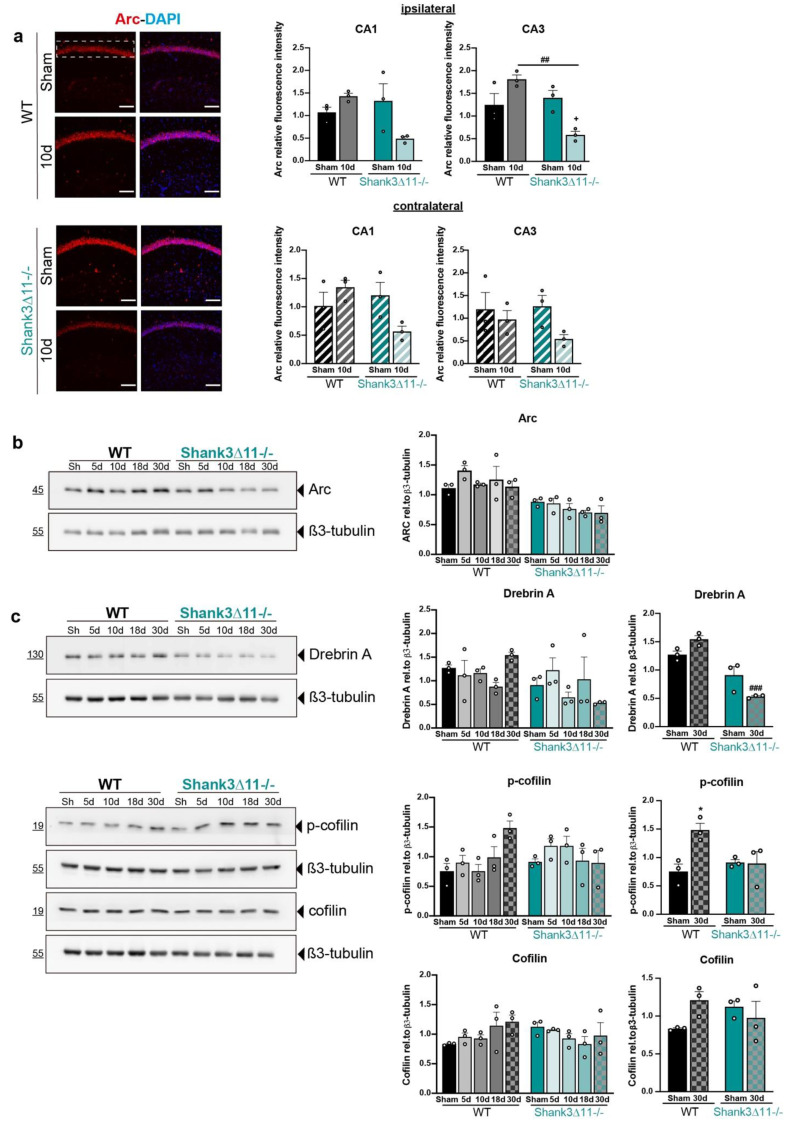
Expression of synaptic and structural plasticity proteins are mainly affected in *Shank3*∆*11−/−* animals after mTBI. (**a**) Immunostaining for Arc protein after 10d of mTBI (left panel). Representative images for the CA1 ipsilateral hippocampal region for WT (upper panel) and *Shank3*∆*11−/−* (bottom panel) mice. The white dashed rectangle represents the region quantified for fluorescence intensity on the ipsilateral side (right upper panel, filled bars) and the contralateral side (right bottom panel, hatched bars) in the CA1 and CA3 hippocampus of WT (black bars) and *Shank3*∆*11−/−* (turquoise bars) mice. N = 3. Datasets were analyzed using two-way ANOVA (genotype × mTBI intervention), and Bonferroni’s post hoc comparison test was performed. [CA1 ipsilateral side: genotype F(1, 8) = 2.887, *p* = 0.1277; mTBI intervention F(1, 8) = 1.411, *p* = 0.2690; interaction genotype × mTBI F(1,8) = 8.719, *p* = 0.0183. CA1 contralateral side: genotype F(1, 8) = 6.913, *p* = 0.0302; mTBI intervention F(1, 8) = 0.4284; interaction genotype × mTBI F(1,8) = 6.913, *p* = 0.0302]. [CA3 ipsilateral side: genotype F(1, 8) = 11.03, *p* = 0.0105; mTBI intervention F(1, 8) = 0.6144, *p* = 0.4557; interaction genotype × mTBI F(1,8) = 18.23, *p* = 0.0027. CA3 contralateral side: genotype F(1, 8) = 0.5416, *p* = 0.4828; interaction genotype × mTBI F(1,8) = 3.651, *p* = 0.0924; interaction genotype × mTBI F(1,8) = 1.012, *p* = 0.3439. (**b**) Western blot for Arc expression in P2 fractions of ipsilateral hippocampus lysates (left panel) and the relative quantification of Arc expression relative to β3-tubulin (right panel) in WT (black bars) and *Shank3*∆*11−/−* (turquoise bars) mice. N = 3. Data are shown as mean ± SEM. Datasets were analyzed using two-way ANOVA (genotype × mTBI intervention), and Bonferroni’s post hoc comparison test was performed. [genotype F(1, 20) = 47.95, *p* < 0.0001; mTBI intervention F(4, 20) = 1.292, *p* = 0.3067; interaction genotype × mTBI F(4, 20) = 0.8788, *p* = 0.4942]. (**c**) Western blot expression analysis of the ipsilateral hippocampal cytosolic compartment (S2 fraction) for drebrin A, p-cofilin and cofilin (left panel) and the quantification of its expression relative to β3-tubulin (right panel) in WT (black bars) and *Shank3*∆*11−/−* (turquoise bars) mice. N = 3. Data are shown as mean ± SEM. Datasets were analyzed using two-way ANOVA (genotype × mTBI intervention), and Bonferroni’s post hoc comparison test was performed. [drebrin A: genotype F(1, 20) = 5.884, *p* = 0.0249; mTBI intervention F(4, 20) = 0.4944, *p* = 0.7400; interaction genotype × mTBI F(4, 20) = 2.580, *p* = 0.0687. Drebrin A at 30 d: genotype F(1, 8) = 61.57, *p* < 0.0001; mTBI intervention F(1, 8) = 0.3714, *p* = 0.5591; interaction genotype × mTBI F(1, 8) = 13.65, *p* = 0.0061]. [p-cofilin: genotype F(1, 20) = 0.2243, *p* = 0.6409; mTBI intervention F(4, 20) = 1.516, *p* = 0.2354; interaction genotype × mTBI F(4, 20) = 3.509, *p* = 0.0252. p-cofilin at 30 d: genotype F(1, 8) = 2.404, *p* = 0.1596; mTBI intervention F(1, 8) = 6.554, *p* = 0.0336; interaction genotype × mTBI F(1, 8) = 7.184, *p* = 0.0279]. [Cofilin: genotype F(1, 20) = 0.1238, *p* = 0.7286; mTBI intervention F(4, 20) = 0.4915, *p* = 0.7420; interaction genotype × mTBI F(4, 20) = 2.026, *p* = 0.1294. Cofilin at 30 d: genotype F(1, 8) = 0.04807, *p* = 0.8319; mTBI intervention F(1, 8) = 0.7919, *p* = 0.3995; interaction genotype × mTBI F(1, 8) = 4.100, *p* = 0.0775]. *: comparison regarding WT sham animals; +: comparison regarding *Shank3*∆*11−/−*, ###: comparison between WT and *Shank3*∆*11−/−* after the same time point of mTBI. * *p* < 0.05: comparison regarding WT sham animals; + *p* < 0.05: comparison regarding *Shank3*∆*11−/−* sham animals; ## *p* < 0.01: comparison between WT sham and *Shank3*∆*11−/−* sham animals.

**Figure 6 ijms-23-06081-f006:**
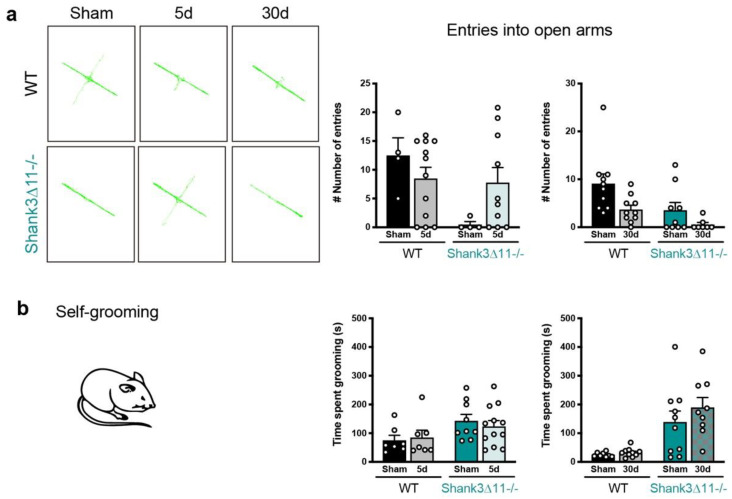
Wild-type animals show no anxiety-like behavior after mTBI, while repetitive behaviors are not exacerbated in *Shank3*∆*11−/−* following mTBI. (**a**) Anxiety assessment through the elevated plus maze (EPM) after 5 d and 30 d of mTBI. Representative tracking images of WT mice trajectory (left upper panel) compared to *Shank3*∆*11−/−* animals (left bottom panel) on the EPM. Quantification of the number of entries into the open arms of the EPM (right panel) of WT (black bars) and *Shank3*∆*11−/−* (turquoise bars). N = 4–12, data are shown as mean ± SEM. Datasets were analyzed using two-way ANOVA (genotype × mTBI intervention), and Bonferroni’s post hoc comparison test was performed. [Entries into the open arm at 5 d: genotype F(1, 26) = 5.074, *p* = 0.0330; mTBI intervention F(1, 26) = 0.3374, *p* = 0.5664; interaction genotype × mTBI F(1, 26) = 3.990, *p* = 0.0563. Entries into the open arm at 30 d: genotype F(1, 32) = 8.457, *p* = 0.0066; mTBI intervention F(1, 32) = 7.903, *p* = 0.0084; interaction genotype × mTBI F(1, 32) = 0.6562, *p* = 0.4239]. (**b**) The time spent performing self-grooming was quantified during 20 min in WT (black bars) compared to *Shank3*∆*11−/−* (turquoise bars) mice after 5 d and 30 d of mTBI. N = 7–12. Data are shown as mean ± SEM. Datasets were analyzed using two-way AN OVA (genotype × mTBI intervention), and Bonferroni’s post hoc comparison test was performed. [Self-grooming at 5 d: genotype F(1, 31) = 5.651, *p* = 0.0238; mTBI intervention F(1, 31) = 0.04366, *p* = 0.8359; interaction genotype × mTBI F(1, 31) = 0.4236, *p* = 0.5199. Self-grooming at 30 d: genotype F(1, 33) = 25.46, *p* < 0.0001; mTBI intervention F(1, 33) = 1.182, *p* = 0.2849; interaction genotype × mTBI F(1, 33) = 0.6416, *p* = 0.4288].

**Figure 7 ijms-23-06081-f007:**
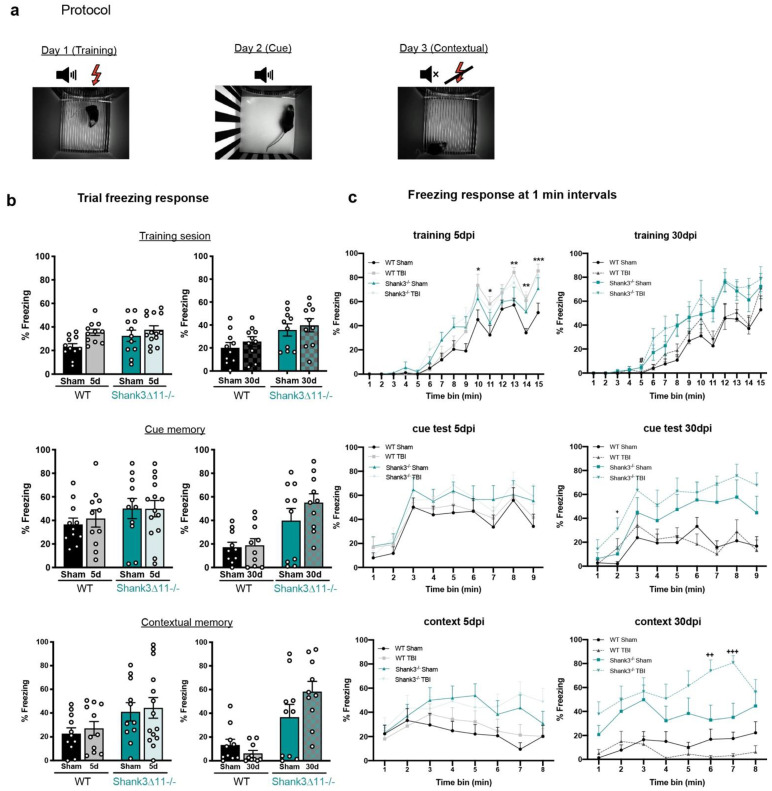
Trace fear conditioning memory was slightly affected in both genotypes following mTBI. (**a**) Depiction of the protocol for the training, auditory or cued memory and contextual fear conditioning test sessions after 5 d and 30 d of mTBI. (**b**) Percentage of the time spent freezing during each whole trial. N = 10–14. Data are shown as mean ± SEM. Datasets were analyzed using two-way ANOVA (genotype × mTBI intervention), and Bonferroni’s post hoc comparison test was performed. [Training session at 5 d: genotype F(1, 43) = 2.723, *p* = 0.1062; mTBI intervention F(1, 43) = 6.122, *p* = 0.0174; interaction genotype × mTBI: F(1, 43) = 1.058, *p* = 0.3093. Training session at 30 d: genotype F(1, 36) = 8.862, *p* = 0.0052; mTBI intervention F(1, 36) = 0.8784, p = 0.3549; interaction genotype × mTBI: F(1, 36) = 0.1934, *p* = 0.8902]. [Cue session at 5 d: genotype F(1, 43) = 2.317, *p* = 0.1353; mTBI intervention F(1, 43) = 0.1131, *p* = 0.7383; interaction genotype × mTBI: F(1, 43) = 0.1293, *p* = 0.7209. Cue session at 30 d: genotype F(1, 36) = 16.38, *p* = 0.0003; mTBI intervention F(1, 36) = 1.353, *p* = 0.2525; interaction genotype × mTBI: F(1, 36) = 0.8815, *p* = 0.3541]. [Contextual session at 5 d: genotype F(1, 43) = 5.895, *p* = 0.0194; mTBI intervention F(1, 36) = 0.2757, *p* = 0.6022; interaction genotype × mTBI: F(1, 43) = 0.0056, *p* = 0.9408]. (**c**) The freezing response was analyzed at 1-min intervals during the different trials in WT (black bars) and *Shank3*∆*11−/−* (turquoise bars) mice. N = 10–14. Data are shown as mean ± SEM. Datasets were analyzed using three-way mixed ANOVA (genotype × mTBI intervention × time) including repetitive measures, and Bonferroni’s post hoc comparison test was performed. [Training at 5 d: genotype × mTBI intervention × time F(2,14) = 1.035, *p* = 0.416; mTBI intervention × time F(1,14) = 3.176, *p* < 0.0001; genotype × time F(1,14) = 1.568, *p* = 0.83. Training at 30 d: genotype × mTBI intervention × time F(2,14) = 0.281, *p* = 0.996; mTBI intervention × time F(1,14) = 0.725, *p* = 0.749; genotype × time F(1,14) = 3.868, *p* < 0.0001]. [Cue test at 5 d: genotype × mTBI intervention × time F(2,8) = 0.464, *p* = 0.881; mTBI intervention × time F(1,8) = 0.440, *p* = 0.897; genotype × time F(1,8) = 0.863, *p* = 0.548. Cue test at 30 d: genotype × mTBI intervention × time F(2,8) = 0.325, *p* = 0.956; mTBI intervention × time F(1,8) = 1.088, *p* = 0.372; genotype × time F(1,8) = 4.804, *p* < 0.0001]. [Contextual test at 5 d: genotype × mTBI intervention × time F(2,7) = 1.186, *p* = 0.311; mTBI intervention × time F(1,7) = 0.500, *p* = 0.835; genotype × time F(1,7) = 2.048, *p* = 0.49. Contextual test at 30 d: genotype × mTBI intervention × time F(2,7) = 3.898, *p* < 0.0001; mTBI intervention × time F(1,7) = 1.385, *p* = 0.212; genotype × time F(1,7) = 1.746, *p* = 0.099]. *: *p* < 0.05, ** *p* < 0.01; *** *p* < 0.001: comparison regarding WT sham animals; ++ *p* < 0.01; +++ *p* < 0.001: comparison regarding *Shank3*∆*11−/−* sham animals.

**Figure 8 ijms-23-06081-f008:**
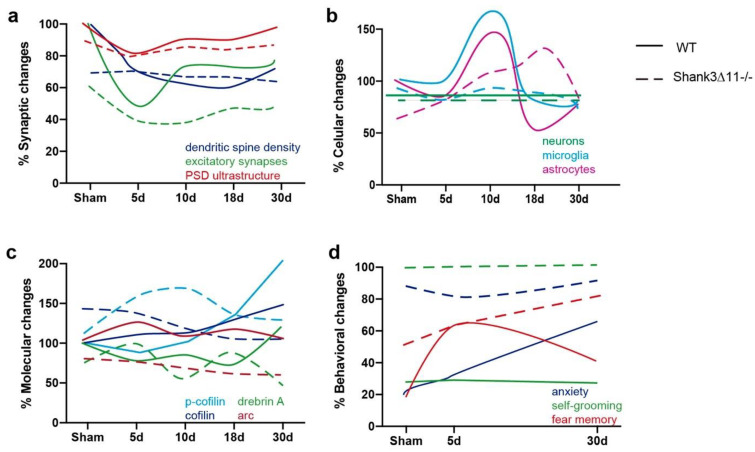
Comparison and time-related changes of the main findings observed in this study between wild-type and *Shank3*∆*11−/−* mice. (**a**) Comparison of the synaptic, (**b**) cellular, (**c**) molecular and (**d**) behavioral differences found between WT and *Shank3*∆*11−/−* before and after the different time points evaluated after mTBI.

## Data Availability

Data are contained within the article or Appendix A. The data presented in this study are available on request from the corresponding author.

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
