# Peer review of "Deletion of the Autism-Associated Protein SHANK3 Abolishes Structural Synaptic Plasticity after Brain Trauma"

_ijms, 2022, doi:10.3390/ijms23116081_

Round 1

Reviewer 1 Report

The subject of the manuscript by Urrutia-Ruiz et al. is the impact of standardised mild traumatic brain injury (mTBI) to WT and Shank3Δ11-/- mice. By comparing morphological changes of hippocampal excitatory synapses the authors show inadequate structural synaptic plasticity in Shank3Δ11-/- mice compared to WT mice. These results were concomitant with molecular markers for synaptic plasticity and could also help to interpret the behavioural test conducted by the authors.

The manuscript is well written and data as well as methods are sound. I have however several minor concerns mainly regarding the design and interpretation of experiments that need to be addressed.

In general:

(1) For future projects it would be great to close the gap between microscopic/molecular methods and behavioural experiments by conducting electrophysiological measurements.

In this context I expected some electrophysiological data (LTP, LTD) by reading the title. May be it’s appropriate to concretise the title by adding the word “structural” or “morphological”?

(2) I appreciate the presented figures that are rich in content. Nevertheless some graphs (especially the labelling) are too small in my point of view. I would kindly ask the authors to check the figures and improve the readability if possible.

(3) In my opinion data that is described and discussed in detail should be placed in the main part of the manuscript (and not in the supplements). With respect to potential restriction of the journal I would suggest the authors to transfer some supplemental figures to the main part of the article.

It would be interesting to know if fear conditioning itself is affecting spine density and PSD ultrastructure Shank3-dependently.

Introduction

(4) Please add at least one reference to the statement in line 70-71 of page 2.

(5) By reading the introduction for me it’s not clear why the authors focused on the hippocampus. Many clinical data regarding Shank3 in the context of ASDs predominantly reveal morphological changes of the cerebral cortex. Furthermore in the context of fear condition the amygdala should be the main region of interest. I would ask the authors to explain why they examined the hippocampus. Please close the loop between Shank3, ASD, fear and hippocampus. May be some references can help to underline the importance of the hippocampus in this context.

Section 2.1.

(6) There seems to be an increase of long thin spines in Shank3Δ11-/- mice after mTBI, which has been neglected by the authors.

Section 2.2.

(7) In line 148-149 of page 5 the authors wrote: “…which recovered close to the baseline levels after 30 dpi on the ipsilateral CA1 hippocampus.” A recovery close to the baseline levels is not supported by the data, since there is a robust statistical difference between sham and 30d animals (as indicated in figure 2b).

Section 2.5.

(8) In line 244 of page 8 the authors wrote: “In contrast, our Shank3Δ11-/- mice seem to not respond to mTBI”. This statement is in contrast to the presented data so far (see figure 1). I would recommend an impartial presentation and interpretation of the data to the authors.

(9) In line 261-262 of page 8 the authors wrote: “Following mTBI, Arc expression decreased after 5 and 18 days in Shank3Δ11-/- compared to WT mice counterparts (Fig. 3b).” This statement suggests an impact of mTBI on Arc expression in Shank3Δ11-/- mice which to my knowledge is not supported by the statistical analysis given in the subtitle of figure 3. By the way I couldn’t find the corresponding statistical values to this statement. Please provide evidence that these differences refer to mTBI and not to genotype.

Section 2.6.

(10) The titles of the section “Shank3Δ11-/- mice do not present behavioral alterations after mTBI” and of the corresponding figure 4 “Wild-type animals are predominantly affected, while Shank3Δ11-/- did not present an altered behavioral response after mTBI.” are not applicable since the authors present statistical differences between Shank3Δ11-/- sham and Shank3Δ11-/- mTBI treated mice (see figure 4c). In my opinion for both groups (WT and Shank3Δ11-/-) the observed effects of mTBI are moderate. I would prefer a more accurate interpretation based on the data.

(11) In line 316-317 of page 11 the authors wrote: “At baseline, Shank3Δ11-/- showed anxiety compared to WT mice.” The comparison of sham WT and sham Shank3Δ11-/- mice reveals a massive avoidance (nearly 0) of entries in the Shank3Δ11-/- group. If the authors hypothesise that mTBI may increase anxiety this behavioural test is inappropriate because Shank3Δ11-/- mice are at the lowest level of a dynamic range. Less entries than nearly 0 are not possible.

(12) In line 322-324 of page 11 the authors wrote: “After five days of mTBI, we observed that WT mice, which spent more time in the center of the arena, are likely to present disorientated (delirium) more than non-anxious behaviour after the injury.” This is highly speculative and raises doubts in the suitability of the selected test and/or time point. In case of a scientific evaluation of sensomotoric deficits due to the surgical procedure a potential “delirium” may occlude an effect of mTBI on Shank3Δ11-/- mice in the open field test.

(13) In line 370-383 of page 13 (figure 4c): For cue and contextual memory there are some sham animals in both groups (but predominantly in the Shank3Δ11-/- group) with nearly 100 % freezing time which cannot be escalated in a statistical relevant manner. In these cases the used stimulation paradigm is insufficient to discover potential increasing effects of mTBI. May be it’s reasonable to exclude data of these specific animals from the analysis of the impact of mTBI?

Author Response

25th May 2022

Deletion of the autism-associated protein SHANK3 abolishes structural synaptic plasticity after brain trauma by Urrutia-Ruiz et al.

Reviewers comments

We would like to express our sincere gratitude to all the Reviewers for their constructive comments and corrections. Thanks to your suggestions and input, the quality of the manuscript has been considerably improved. Please, find below our point-by-point rebuttal; our replies are in blue.

Referee #1 (Comments and Suggestions for Authors)

The subject of the manuscript by Urrutia-Ruiz et al. is the impact of standardised mild traumatic brain injury (mTBI) to WT and Shank3Δ11-/- mice. By comparing morphological changes of hippocampal excitatory synapses, the authors show inadequate structural synaptic plasticity in Shank3Δ11-/- mice compared to WT mice. These results were concomitant with molecular markers for synaptic plasticity and could also help to interpret the behavioural test conducted by the authors.

The manuscript is well written and data as well as methods are sound. I have however several minor concerns mainly regarding the design and interpretation of experiments that need to be addressed.

In general:

(1) For future projects it would be great to close the gap between microscopic/molecular methods and behavioural experiments by conducting electrophysiological measurements.

In this context I expected some electrophysiological data (LTP, LTD) by reading the title. May be it’s appropriate to concretise the title by adding the word “structural” or “morphological”?

We have now changed the article’s title to: “Deletion of autism-associated protein SHANK3 abolishes structural synaptic plasticity after brain trauma.”

(2) I appreciate the presented figures that are rich in content. Nevertheless some graphs (especially the labelling) are too small in my point of view. I would kindly ask the authors to check the figures and improve the readability if possible.

We increased the font size of all the main and supplementary figures. We additionally split the main behavior figure to improve its readability.

(3) In my opinion data that is described and discussed in detail should be placed in the main part of the manuscript (and not in the supplements). With respect to potential restriction of the journal I would suggest the authors to transfer some supplemental figures to the main part of the article.

It would be interesting to know if fear conditioning itself is affecting spine density and PSD ultrastructure Shank3-dependently.

We agree with Referee 1 and will move the information related to the cellular changes concerning neurons, astrocytes and microglia that we found in both genotypes following mTBI. Also, whether fear conditioning itself could alter the spine density and ultrastructure is worth analyzing and considering doing in our future research.

Introduction

(4) Please add at least one reference to the statement in line 70-71 of page 2.

We added the paper of Chen et al. They used a maternal immune activation (MIA) model (injection of poly[I:C] to dam at gestational day 12.5) along with perinatal hypoxia-ischemia insult at postnatal day 10. Both interventions added autistic-like features to the MIA model (line 75) [1].

(5) By reading the introduction for me it’s not clear why the authors focused on the hippocampus. Many clinical data regarding Shank3 in the context of ASDs predominantly reveal morphological changes of the cerebral cortex. Furthermore, in the context of fear condition the amygdala should be the main region of interest. I would ask the authors to explain why they examined the hippocampus. Please close the loop between Shank3, ASD, fear and hippocampus. May be some references can help to underline the importance of the hippocampus in this context.

We focused on the hippocampus because it is involved in learning and memory, especially the CA1 hippocampus, which is crucial for converting new memories into long-term memories (consolidation). Moreover, in developing hippocampal neurons, SHANK3 is expressed in axons and presynaptic specializations and modulates the expression of the NMDA receptors at the axon terminal [2]. In vivo, SHANK3, apart from being expressed throughout the brain, is enriched in the hippocampus [3]. At the functional level, Shank3 knockout animals have shown impairments in hippocampal LTP [4-6]. On the other hand, it is known that individuals with a mild TBI show cognitive impairments without the presence of a distinguishable injury [7]. We performed a trace-fear conditioning protocol intended to measure hippocampal-dependent function to assess hippocampal function. Unlike the delay-fear conditioning protocol, animals with hippocampal lesions could not associate the conditioned with the unconditioned stimulus in the presence of a brief empty trace interval. In the trace-fear conditioning paradigm, the hippocampus is selectively involved rather than a general fear learning or expression [8]. Even though there are controversial findings regarding the amygdala’s involvement in this behavior, inactivation of this structure with the GABAA agonist muscimol revealed impairment in contextual and delay conditioning, but not in the acquisition and consolidation of trace fear conditioning [9]. Moreover, in previous TBI analyses synapses of the hippocampus have been used to analyze structural plasticity. Therefore the hippocampus was an important brain region to relate our findings to published data.

The introduction provided now information on the loop between Shank3, ASD, fear, and the hippocampus (lines 84-89)

Section 2.1.

(6) There seems to be an increase of long thin spines in Shank3Δ11-/- mice after mTBI, which has been neglected by the authors.

The statistically significant increase in the number of long-thin spines in Shank3 knockout animals following mTBI was mentioned in lines 120-121 and discussed in 1091-1092.

Section 2.2.

(7) In line 148-149 of page 5 the authors wrote: “…which recovered close to the baseline levels after 30 dpi on the ipsilateral CA1 hippocampus.” A recovery close to the baseline levels is not supported by the data, since there is a robust statistical difference between sham and 30d animals (as indicated in figure 2b).

We referred to the changes observed regarding the fine ultrastructure of the PSD. However, we added that we did not see a recovery in the overall number of PSD structures in line 171 for Shank3∆11-/- mice.

Section 2.5.

(8) In line 244 of page 8 the authors wrote: “In contrast, our Shank3Δ11-/- mice seem to not respond to mTBI”. This statement is in contrast to the presented data so far (see figure 1). I would recommend an impartial presentation and interpretation of the data to the authors.

We do agree with Reviewer 1 and changed our conclusion to “our Shank3∆11-/- mice did not lose excitatory synapses and showed perturbed dendritic spine structural remodeling following mTBI. These findings led us to hypothesize that Shank3∆11-/- mice display impairments in morphological plasticity” (line 404-406).

(9) In line 261-262 of page 8 the authors wrote: “Following mTBI, Arc expression decreased after 5 and 18 days in Shank3Δ11-/- compared to WT mice counterparts (Fig. 3b).” This statement suggests an impact of mTBI on Arc expression in Shank3Δ11-/- mice which to my knowledge is not supported by the statistical analysis given in the subtitle of figure 3. By the way I couldn’t find the corresponding statistical values to this statement. Please provide evidence that these differences refer to mTBI and not to genotype.

We only observed an interaction effect of the genotype and the mTBI intervention in the immunostainings for the CA1 and CA3 regions [10]. As been said, in western blots, we only found a difference in the genotype that does not enable us to conclude that there was a decrease in Arc’s expression due to the genotype x mTBI interaction. Therefore, we deleted this information from the manuscript in line 426.

Section 2.6.

(10) The titles of the section “Shank3Δ11-/- mice do not present behavioral alterations after mTBI” and of the corresponding figure 4 “Wild-type animals are predominantly affected, while Shank3Δ11-/- did not present an altered behavioral response after mTBI.” are not applicable since the authors present statistical differences between Shank3Δ11-/- sham and Shank3Δ11-/- mTBI treated mice (see figure 4c). In my opinion for both groups (WT and Shank3Δ11-/-) the observed effects of mTBI are moderate. I would prefer a more accurate interpretation based on the data.

We agree with Reviewer 1 that the statistical analysis enables us to conclude that Shank3∆11-/- mice had an increase in freezing behavior in the contextual test following 30 days of mTBI (interaction genotype x mTBI x time: p < 0.0001) (Fig. 7c, bottom panel). Therefore, we modified the title: “Wild-type and Shank3Δ11-/- presented moderate alterations after mTBI” (Line 488). We also added in the introduction that we observed subtle alterations due to mTBI on both genotypes (Line 91-92).

(11) In line 316-317 of page 11 the authors wrote: “At baseline, Shank3Δ11-/- showed anxiety compared to WT mice.” The comparison of sham WT and sham Shank3Δ11-/- mice reveals a massive avoidance (nearly 0) of entries in the Shank3Δ11-/- group. If the authors hypothesise that mTBI may increase anxiety this behavioural test is inappropriate because Shank3Δ11-/- mice are at the lowest level of a dynamic range. Less entries than nearly 0 are not possible.

Regarding Shank3∆11-/- mice, we acknowledge that these animals at baseline already present an anxious behavior with no fewer entries as zero. Thus, we also conducted the open field test, and here at 5 dpi, we only observed changes in the time spent on the center or borders of the arena in WT mice following mTBI, while Shank3-deficient mice did not present changes due mTBI (genotype x mTBI: p = 0.0199).

We added that mTBI did not induce anxiety in WT mice after 30 days (interaction between genotype x mTBI: p = 0.4239), only a trend (line 495).

(12) In line 322-324 of page 11 the authors wrote: “After five days of mTBI, we observed that WT mice, which spent more time in the center of the arena, are likely to present disorientated (delirium) more than non-anxious behaviour after the injury.” This is highly speculative and raises doubts in the suitability of the selected test and/or time point. In case of a scientific evaluation of sensomotoric deficits due to the surgical procedure a potential “delirium” may occlude an effect of mTBI on Shank3Δ11-/- mice in the open field test.

We deleted this conclusion; however, we do not have an explanation for WT animals expending more time on the center of the arena following mTBI at five dpi (line 500-501).

(13) In line 370-383 of page 13 (figure 4c): For cue and contextual memory there are some sham animals in both groups (but predominantly in the Shank3Δ11-/- group) with nearly 100 % freezing time which cannot be escalated in a statistical relevant manner. In these cases the used stimulation paradigm is insufficient to discover potential increasing effects of mTBI. May be it’s reasonable to exclude data of these specific animals from the analysis of the impact of mTBI?

Initially, we analyzed the freezing behavior when the tone was presented in the training and cue sessions (this we forgot to mention it in the methods section), and in doing so, we had animals freezing 100% of the tone time. We corrected this issue and added the information for the % of freezing in the whole trial (for the training and cue sessions.  Now, none of the animals are freezing 100% of the whole trial time (see figure 7). Additionally, we added the new statistical analysis to the figure legend. After doing these changes, the statistical analysis is corrected accordingly.

Reviewer 2 Report

This well designed, executed and written preclinical study contributes to the filed of ASD as it found that SHANK3 is essential to induce synaptic plasticity after a mTBI confirming that Shank3∆11mice present plasticity impairments and behavioral inflexibility. Additionally, some  of  the  proteins  known  to  mediate  crucial  structural  and  functional plasticity  changes  following  mTBI  are dysregulated  in Shank3-deficient  mice.  This is the first time that show no apparent synaptic response to a physical brain impact induced by brain trauma that has been inflicted on a mice model of autism. In summary, the study  found  good  evidence  that SHANK3 isoforms  are  essential  for plastic  changes  of  excitatory  synapses. In  turn, Shank3 knockout mice  completely lack structural synaptic plasticity, which might at least in part explain the rigidity of behaviors, problems in adjusting to new situations and cognitive deficits seen in ASDs.

A minor suggestion:
“We found that Shank3∆11-/-mice have fewer dendritic spines with altered morphology, fewer excitatory synapses, and smaller PSDs (Fig. 1, 2). These changes indicate profound changes at the  synapse, where  SHANK3 is predominantly located [6].”
This is all fine. Could the authors contrast an established dendritic abnormalities findings in fragile X syndrome mouse model https://doi.org/10.1038/s41467-019-11891-6
And in humans with FXS 10.1093/cercor/10.10.1038 In short, the Fragile X Mental Retardation Protein, which lacks in fragile X syndrome (FXS) and is associated with severity of the neurobehavioral phenotype in FXS  https://doi.org/10.3390/brainsci10100694, appears to be essential to processes necessary for the maturation and pruning of dendritic spines, either during development or throughout the lifetime of an organism.

Author Response

25th May 2022

Deletion of the autism-associated protein SHANK3 abolishes structural synaptic plasticity after brain trauma by Urrutia-Ruiz et al.

Reviewers comments

We would like to express our sincere gratitude to all the Reviewers for their constructive comments and corrections. Thanks to your suggestions and input, the quality of the manuscript has been considerably improved. Please, find below our point-by-point rebuttal; our replies are in blue.

Referee #2 (Comments and Suggestions for Authors)

This well designed, executed and written preclinical study contributes to the field of ASD as it found that SHANK3 is essential to induce synaptic plasticity after a mTBI confirming that Shank3∆11 mice present plasticity impairments and behavioral inflexibility. Additionally, some of the proteins known to mediate crucial structural and functional plasticity changes following mTBI are dysregulated in Shank3-deficient mice. This is the first time that show no apparent synaptic response to a physical brain impact induced by brain trauma that has been inflicted on a mice model of autism. In summary, the study found good evidence that SHANK3 isoforms are essential for plastic changes of excitatory synapses. In turn, Shank3 knockout mice completely lack structural synaptic plasticity, which might at least in part explain the rigidity of behaviors, problems in adjusting to new situations and cognitive deficits seen in ASDs.

A minor suggestion:

“We found that Shank3∆11-/-mice have fewer dendritic spines with altered morphology, fewer excitatory synapses, and smaller PSDs (Fig. 1, 2). These changes indicate profound changes at the synapse, where SHANK3 is predominantly located [6].”This is all fine. Could the authors contrast an established dendritic abnormalities findings in fragile X syndrome mouse model https://doi.org/10.1038/s41467-019-11891-6And in humans with FXS 10.1093/cercor/10.10.1038 In short, the Fragile X Mental Retardation Protein, which lacks in fragile X syndrome (FXS) and is associated with severity of the neurobehavioral phenotype in FXS  https://doi.org/10.3390/brainsci10100694, appears to be essential to processes necessary for the maturation and pruning of dendritic spines, either during development or throughout the lifetime of an organism.

We incorporated this suggestion into our discussion in lines 1230-1241. Mentioning that: “ In fact, plasticity impairments have also been observed in the case of fragile X syndrome, in which the loss of FMRP protein, RNA binding protein that regulates translation and long-term changes in synaptic strength, leads to intellectual disability and, in some cases to ASD [11]. Like Shank3∆11-/-, these animals present dendritic spines with an immature phenotype; however, in humans, there is an increase in dendritic spine density due to pruning defects, a phenotype that has not been consistently replicated in knockout mice [12]. These animals present a disrupted synapse formation by showing multiple innervated spines leading to abnormal synaptogenesis [13], and in humans, the severity of the de syndrome in terms of intellectual disability is inversely correlated with FMRP expression levels [14]. Within SHANK mutations in humans, SHANK3 mutations have been related to more severe intellectual disability than SHANK1 and SHANK2 [15].

Reviewer 3 Report

The study of Carolina Urrutia-Ruiz and co-authors is very exciting and points to SHANK3 protein as a key regulator synaptic plasticity and, possibly, neuroinflammation in response to mild TBI. Since the SHANK3 gene is associated with autism spectrum disorders (ASD), the data obtained can explain “unresponsively” of the nerve system in case of some forms of inherent ASD. The paper is nicely presented in figures and relies on solid data. So, I have only minor recommendations. Please, see below.

Introduction:

  1. Synaptic changes in ASD occur at both postsynaptic and presynaptic side. Could authors add information about possible presynaptic alteration in Shank KO-mice?
  2. Is there information about changes in GABAergic transmission in Shank KO mice?

Results:

  1. Authors precisely estimated PSD metrics in TEM microphotographs. Do the authors consider it necessary to additionally estimate the size of total synaptic vesicle pool and docking vesicle number per excitatory synapse? Also, the mitochondrial quantifications in synapses can give some interesting bits of data.
  2. About Arc. The protein has two main residences in postsynaptic region and nucleus. Can the authors compare the changes in Arc abundance in nucleus region based on IHC data showed in the manuscript? Probably it will be helpful to detect the translocation of Arc.

Methods:

  1. How was the quality of crude synaptosome fraction assessed?

Author Response

25th May 2022

Deletion of the autism-associated protein SHANK3 abolishes structural synaptic plasticity after brain trauma by Urrutia-Ruiz et al.

Reviewers comments

We would like to express our sincere gratitude to all the Reviewers for their constructive comments and corrections. Thanks to your suggestions and input, the quality of the manuscript has been considerably improved. Please, find below our point-by-point rebuttal; our replies are in blue.

Referee #3 (Comments and Suggestions for Authors)

The study of Carolina Urrutia-Ruiz and co-authors is very exciting and points to SHANK3 protein as a key regulator synaptic plasticity and, possibly, neuroinflammation in response to mild TBI. Since the SHANK3 gene is associated with autism spectrum disorders (ASD), the data obtained can explain “unresponsively” of the nerve system in case of some forms of inherent ASD. The paper is nicely presented in figures and relies on solid data. So, I have only minor recommendations. Please, see below.

Introduction:

  1. Synaptic changes in ASD occur at both postsynaptic and presynaptic side. Could authors add information about possible presynaptic alteration in Shank KO-mice?

This is an interesting point brought up by the reviewer. We mentioned the effect of Shank3 deletion on the presynaptic side in lines 65-69: “Opposite results have been observed regarding the effects of Shank3 deletion on the presynaptic side, where SHANK3 is also present [2]. For instance, a reduced presynaptic release in D2-medium spiny neurons [16] or increased presynaptic release in Shank3 heterozygous mice [5], or no changes in the hippocampal presynaptic release in Shank3 knockout mice [17,18]”. We did not see obvious structural alterations in the presynaptic terminals and –as proof of principle we screened for changes in presynaptic mitochondrial number and morphology as well as for the alterations of the number of docked vesicles but could not find significant alterations. The data sets are, however, to immature to be included in the manuscript.

  1. Is there information about changes in GABAergic transmission in Shank KO mice?

Indeed. We added this information to our introduction (in lines 69-74): “SHANK3 is not only expressed in excitatory but in inhibitory neurons. GABAergic neurons in the prefrontal cortex and dorsolateral striatum express SHANK3. SHANK3 specific deletion in GABAergic neurons suppressed excitatory transmission without impacting inhibitory transmission; additionally, the abnormal social and locomotor behavior of the global Shank3 knockout mice in this study was replicated in SHANK3-GABAergic knockout mice” [19].

Results:

  1. Authors precisely estimated PSD metrics in TEM microphotographs. Do the authors consider it necessary to additionally estimate the size of total synaptic vesicle pool and docking vesicle number per excitatory synapse? Also, the mitochondrial quantifications in synapses can give some interesting bits of data.

We did not consider analyzing presynaptic changes in depth. Instead, we focused mainly on the postsynaptic side, where SHANK3 is highly enriched. However, the Reviewer is correct. It is also important to focus on this in the future. (see also the answer to point 1)

  1. About Arc. The protein has two main residences in postsynaptic region and nucleus. Can the authors compare the changes in Arc abundance in nucleus region based on IHC data showed in the manuscript? Probably it will be helpful to detect the translocation of Arc.

We closely analyzed the localization of Arc in the different experimental groups but could not detect significant differences in the localization of the Arc protein. We believe that the relatively late time point of analysis (as also discussed in the manuscript) is not optimal to see dynamic changes of localization of this immediate early gene.

Methods:

  1. How was the quality of crude synaptosome fraction assessed?

To assess the quality of the crude synaptosomal fraction, we evaluated the expression of SHANK3 and PSD-95 (which were enriched in the crude synaptosomal (P2) and postsynaptic density (P3) pellet), along with the expression of synaptophysin (which was enriched in the post-nuclear (S1), cytosolic compartment (S2) and synaptic cytosol (S3) supernatant). In addition, we provided a representative western blot of the quality assessment of the hippocampal fractions used for sham animals in this study in supplementary figure 2b. We also added this information to section 4.7 of material and methods (lines 1391-1393)

Reviewer 4 Report

In the present manuscript, the authors used the mild traumatic brain injury (mTBI) paradigm to investigated the synaptic plasticity of excitatory synapses in the CA1 hippocampal region of an animal model knockout for SHANK3, a high-risk autism candidate gene. In Shank3Δ11-/- mice, which showed lesser dendritic spines at baseline, mTBI trigger 1) no significant synaptic alterations, 2) no spine morphology changes, 3) no microglia proliferation, 4) no upregulation of Arc and p-cofilin proteins and 5) no changes in anxiety or fear memory, compared WT animals. 

Overall, the study is extremely original and interesting and also well designed. The manuscript is well written and methods, data and conclusions are clearly presented. 
The authors should the minor issues below.

  1. The authors should add the minus sign in “mean + SEM” in figure 1 and 2.
  2. In paragraph 2.2 the same sentence is repeated 2 times (Line 187-188: “We did not observe significant statistically ultrastructural changes after mTBI in WT or Shank3Δ11-/- mice on the contralateral side”. Line 192-193: “Moreover, we did not find statistically significant changes at the synaptic ultrastructural level in WT mice or Shank3Δ11-/- mice on the contralateral side following mTBI.”)
  3. For the analysis in crude synaptosomes, conducted by western blot, does each band correspond to a single animal or were multiple animals pulled together?

Author Response

25th May 2022

Deletion of the autism-associated protein SHANK3 abolishes structural synaptic plasticity after brain trauma by Urrutia-Ruiz et al.

Reviewers comments

We would like to express our sincere gratitude to all the Reviewers for their constructive comments and corrections. Thanks to your suggestions and input, the quality of the manuscript has been considerably improved. Please, find below our point-by-point rebuttal; our replies are in blue.

Referee 4 (Comments and Suggestions for Authors)

In the present manuscript, the authors used the mild traumatic brain injury (mTBI) paradigm to investigate the synaptic plasticity of excitatory synapses in the CA1 hippocampal region of an animal model knockout for SHANK3, a high-risk autism candidate gene. In Shank3Δ11-/- mice, which showed lesser dendritic spines at baseline, mTBI trigger 1) no significant synaptic alterations, 2) no spine morphology changes, 3) no microglia proliferation, 4) no upregulation of Arc and p-cofilin proteins and 5) no changes in anxiety or fear memory, compared WT animals. 

Overall, the study is extremely original and interesting and also well designed. The manuscript is well written, and methods, data and conclusions are clearly presented.  The authors should the minor issues below.

  1. The authors should add the minus sign in “mean + SEM” in figure 1 and 2.

We corrected this mistake and added the minus sign to the legends of Figures 1 and 2. 

  1. In paragraph 2.2 the same sentence is repeated 2 times (Line 187-188: “We did not observe significant statistically ultrastructural changes after mTBI in WT or Shank3Δ11-/- mice on the contralateral side”. Line 192-193: “Moreover, we did not find statistically significant changes at the synaptic ultrastructural level in WT mice or Shank3Δ11-/- mice on the contralateral side following mTBI.”)

We corrected this mistake and mentioned this result at the end of paragraph 2.2.

  1. For the analysis in crude synaptosomes, conducted by western blot, does each band correspond to a single animal or were multiple animals pulled together?

Each animal corresponds to a single band. The western blot for each animal replicates used in this study is provided in supplementary figures 5 and 6.